# ESLM: RISK-AVERSE SELECTIVE LANGUAGE MODELING WITH HIERARCHICAL BATCH SELECTION

## ABSTRACT

Large language model pretraining is compute-intensive, yet many tokens contribute marginally to learning, resulting in inefficiency. We introduce Efficient Selective Language Modeling (ESLM), an online, risk-aware batch selection algorithm that improves training efficiency and distributional robustness. ESLM operates in two phases: $(i)$ instance-level selection via a shallow early-exit model pass that computes proxy per-instance statistics (e.g., loss or entropy) and retains data points using value-at-risk thresholding; and $(ii)$ loss shaping with token-level selection via risk-aware thresholding on per-token scores. This data-centric mechanism reshapes the training objective, prioritizing high-risk tokens and eliminating redundant gradient computation. We frame ESLM as a bilevel game: the model competes with a masking adversary that selects worst-case token subsets under a constrained thresholding rule. In the loss-based setting, ESLM recovers conditional value-at-risk loss minimization, linking selective pretraining to distributionally robust optimization. We extend our approach to ADA-ESLM, which adaptively tunes the selection confidence during training. Experiments on GPT-2 pretraining show that ESLM significantly reduces training FLOPs while maintaining or improving perplexity and downstream performance compared to baselines. Our approach also scales across model sizes, pretraining corpora, and integrates naturally with knowledge distillation.

## 1 INTRODUCTION

The growing scale of large language models (LLMs) has brought substantial improvements in downstream performance at the expense of significantly higher pretraining costs (Brown et al., 2020). Training LLMs is notoriously compute-intensive, requiring massive GPU resources and often processing billions of tokens uniformly. Yet, many tokens, e.g., predictable or low-entropy ones, contribute little to model learning (Hüllermeier and Waegeman, 2021). Standard causal language modeling (CLM) treats all tokens equally in the loss, allocating compute uniformly to frequent or trivial tokens and more informative ones, leading to inefficient training and suboptimal use of resources (Lin et al., 2024).

Efforts to improve pretraining efficiency span architectural advances (Dao, 2023), token pruning (Hou et al., 2022), and increasingly, data-centric strategies (Xia et al., 2024; Wang et al., 2024). Among these, data-centric approaches show particular promise for sample efficiency through selective weighting or filtering of training examples (Katharopoulos and Fleuret, 2018). However, existing methods often rely on reference models or heuristics (Lin et al., 2024), operate at the sequence level (Yu et al., 2024), or require offline scoring (Xie et al., 2023b; Wettig et al., 2024). These choices limit adaptability and scaling of the model to massive web-scale corpora, and the ability to exploit token-level heterogeneity, despite being crucial for optimizing training dynamics and resource usage in LLM pretraining.

We address this gap with ESLM—*Efficient Selective Language Modeling*— a self-supervised data-centric framework that performs *online token-level batch selection* for efficient and robust pretraining. ESLM proceeds in two phases (see Figure 1): $(i)$ a *proxy phase* that selects instances (sequences) via a shallow early-exit pass using only $L$ model layers based on proxy statistics (e.g., per-instance loss or predictive entropy (Shannon, 1948)), keeping instances by (conditional-)value-at-risk (CVaR/VaR) thresholding; $(ii)$ the training phase with risk-aware *token-level loss shaping*, which retains only the highest-risk tokens in the training objective via VaR thresholding over per-token statistics. This dynamic filtering shapes the training loss to emphasize uncertain or informative tokens, reducing redundant gradient updates and improving compute efficiency.

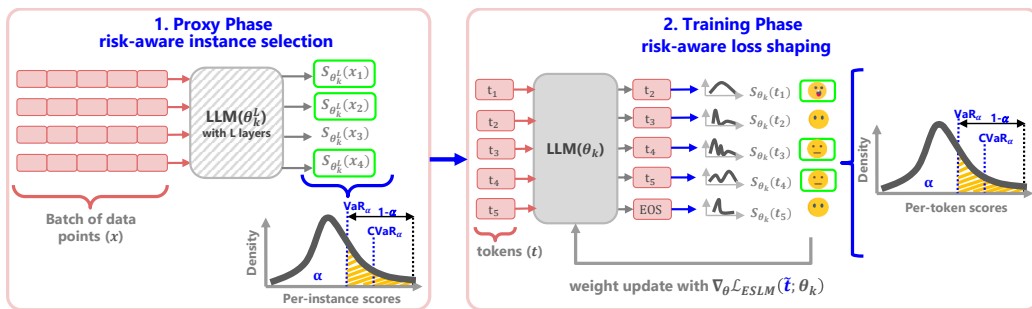

Figure 1: **ESLM illustration.** ESLM uses VaR thresholding on proxy per-instance scores to select high-risk data points, then applies token-level VaR to perform loss shaping. This reshapes the effective training distribution and loss by focusing computational resources on tokens with higher learning value.

Beyond its algorithmic simplicity, ESLM is grounded in a solid theoretical foundation. Its risk-aware selection mechanism can be viewed as a bilevel game in which the model competes with a constrained adversary that restricts learning to the most challenging tokens, directly linked to distributionally robust optimization through targeted reshaping of the training distribution. Building on this, we extend our approach to ADA-ESLM, an adaptive variant that dynamically calibrates the confidence level in response to the training dynamics, enabling a principled control over the compute-efficiency and generalization trade-off. Notably, ESLM requires no auxiliary supervision, reference models, or computationally expensive offline scoring. It also integrates naturally with knowledge distillation, allowing for risk-aware teacher supervision at the token level. Our key contributions are as follows:

- We propose ESLM, a risk-averse, two-phased selective language modeling that prioritizes high-risk inputs (i.e., informative or uncertain) for efficient LLM pretraining. ESLM first performs instance-level filtering using a lightweight proxy, then applies token-level loss shaping on selected instances. Selection in both phases uses VaR thresholding over loss or entropy statistics. We provide its two variants: VaR-entropy and CVaR-loss based on the risk score choice.
- We frame ESLM as a bilevel adversarial game between the model and a masker that perturbs the effective training distribution by selecting worst-case token subsets under a constrained thresholding rule. We further show that for loss-based selection, ESLM admits a distributionally robust optimization interpretation by recovering the CVaR objective (Rockafellar and Uryasev, 2002), a well-established risk-sensitive formulation from robust statistics (Ben-Tal et al., 2009).
- We propose ADA-ESLM, which adaptively adjusts the selection confidence level via a risk-aware controller guided by CVaR feedback to balance compute-efficiency and generalization.
- We demonstrate ESLM's utility for knowledge distillation by enabling a sparsified risk-aware teacher supervision provided for selected high-risk tokens.
- Our experiments on GPT-2 (124M–1.5B) pretraining demonstrate that ESLM significantly reduces training FLOPs while maintaining or improving both validation perplexity and downstream task accuracy, with consistent gains across model sizes, dataset mixtures, and training settings.

## 2 BACKGROUND

This section introduces the key components underlying ESLM: CLM as the pretraining objective, token-level uncertainty estimation, and risk measures: VaR and CVaR.

**Causal Language Modeling (CLM).** CLM trains a language model (LM) $\theta$ to predict each token in a sequence given the previous context. Given a corpus $\mathcal{C}$ of sequences $x = (x_1, \ldots, x_M)$, each is of length $T$, drawn from a distribution $\mathcal{D}$ over a vocabulary $\mathcal{V}$, the model factorizes the joint probability as: $P_\theta(x) = \prod_{j=1}^{T} P_\theta(x_j \mid x_{<j})$. CLM minimizes the average autoregressive loss:

$$\mathcal{L}_{\text{CLM}}(x; \theta) = \mathbb{E}_{x \sim \mathcal{D}}[\ell_\theta(x)] = \frac{1}{T} \sum_{j=1}^{T} -\log P_\theta(x_j \mid x_{<j}),$$

treating all tokens equally, despite many offering a limited learning signal (Lin et al., 2024).

**Token-level risk.** Let $S_\theta(x_j) \in \mathbb{R}$ denote the *risk score* of token $x_j$, under LM $\theta$, computed via:

- (i) **Entropy** (Shannon, 1948): $H_\theta(x_j) = -\sum_{v \in \mathcal{V}} P_\theta(v \mid x_{<j}) \log P_\theta(v \mid x_{<j})$.
- (ii) **Loss**: $\ell_\theta(x_j) = -\log P_\theta(x_j \mid x_{<j})$.

---

**Algorithm 1** ESLM

---

1: **Input:** LM $\theta$, dataset $\mathcal{D}$, learning rate $\eta$, confidence $\alpha \in (0,1)$, batch size $M$, proxy $L$ layers.
2: **for** each training iteration $k = 1, \ldots, K$ **do**
3:      Sample a batch of instances $\mathcal{B} = \{x_1, \ldots, x_M\} \sim \mathcal{D}$ of length $T$ $\{x_j^t\}_{t=1}^T, \forall j \in \{1, ..., M\}$.
4:      Compute per-instance scores $S_{\theta_k^L}(x)$ using early-exit $\theta_k^L$ proxy.      *\/ Entropy or loss
5:      Compute threshold $S_{\theta_k^L, \alpha}^{\text{VaR}} \leftarrow \text{VaR}_\alpha \left( \{S_{\theta_k^L}(x_j)\}_{j=1}^M \right)$ using (1).
6:      $\bar{\mathcal{B}} \leftarrow \{x_j \in \mathcal{B} \mid S_{\theta_k^L}(x_j) \geq S_{\theta_k^L, \alpha}^{\text{VaR}}\}$.      *\/ High-risk instance selection
7:      Compute per-token statistics via: $S_{\theta_k}(x_j^t) = \begin{cases} H_{\theta_k}(x_j^t) \text{ as in (i)}, & \text{(VaR-entropy)} \\ \ell_{\theta_k}(x_j^t) \text{ as in (ii)}, & \text{(CVaR-loss)} \end{cases}$
8:      Compute threshold $S_{\theta_k, \alpha}^{\text{VaR}} \leftarrow \text{VaR}_\alpha \left( \{\{S_{\theta_k}(x_j^t)\}_{j \in \bar{\mathcal{B}}}\}_{t=1}^T \right)$ using (1).
9:      $\tilde{\mathcal{B}} \leftarrow \{x_j^t \in \bar{\mathcal{B}} \mid S_{\theta_k}(x_j^t) \geq S_{\theta_k, \alpha}^{\text{VaR}}\}$.      *\/ High-risk token selection
10:      Compute loss over selected tokens:      *\/ Shaped loss
     $\mathcal{L}_{\tilde{\mathcal{B}}}(x; \theta_k) = \begin{cases} \mathbb{E}[\ell_{\theta_k}(x_j^t) \mid x_j^t \in \tilde{\mathcal{B}}], & \text{(VaR-entropy)} \\ \text{CVaR}_\alpha(\ell_{\theta_k}(x)) = \mathbb{E}[\ell_{\theta_k}(x_j^t) \mid x_j^t \in \tilde{\mathcal{B}}] \text{ using (2)}, & \text{(CVaR-loss)} \end{cases}$
11:      Update model parameters using optimizer $O$: $\theta_{k+1} \leftarrow O(\theta_k, \nabla_\theta \mathcal{L}_{\tilde{\mathcal{B}}}(x; \theta_k), \eta)$.
12: **end for**
13: **return** $\theta_K$.

---

Both measures serve as proxies for token difficulty and informativeness—highlighting ambiguous, uncertain, or mispredicted tokens. **Instance-level risk scores** are obtained by aggregating token-level risk scores, i.e., the mean score over non-padded tokens within each sequence (line 4, Algorithm 1).

**Risk measures.** To prioritize high-impact tokens, we adopt risk-sensitive criteria from robust statistics (Gagne and Dayan, 2021). Let $S_\theta(x_j)$ denote per-token risk score computed via LM $\theta$. The *value-at-risk* (VaR) (Rockafellar et al., 2000) at confidence level $\alpha \in (0,1)$ is the minimum threshold such that only the top $(1 - \alpha)$ fraction of scores exceed the threshold; see Figure 1:

$$\text{VaR}_\alpha(S_\theta) := \inf\{\tau \in \mathbb{R} \mid P(S_\theta \geq \tau) \leq 1 - \alpha\}. \tag{1}$$

The corresponding CVaR is a coherent risk measure (Artzner et al., 1999) that computes the expected score among these highest-risk tokens (Rockafellar and Uryasev, 2002):

$$\text{CVaR}_\alpha(S_\theta) := \min_\tau \mathbb{E}_{x \sim \mathcal{D}} \left[ \tau + \frac{1}{1 - \alpha} \max(0, S_\theta(x) - \tau) \right]. \tag{2}$$

These tail-risk measures allow us to reshape the training objective to emphasize tokens that are difficult or uncertain—an idea we exploit in the ESLM framework for efficient pretraining.

## 3 ESLM: RISK-AVERSE SELECTIVE LANGUAGE MODELING

We now introduce ESLM, a two-phase selective language modeling framework that improves pretraining efficiency by focusing optimization on high-risk inputs. We consider the standard causal language modeling setup presented in Section 2, where a language model with parameters $\theta$ is trained to minimize the expected token-level autoregressive loss. While effective, the expectation-based CLM objective assumes uniform importance across all tokens, leading to two key inefficiencies:

1. It wastes computation on trivially predictable tokens that dominate the loss landscape but offer little training signal.
2. It disregards token-level risk and overlooks rare, ambiguous, or out-of-distribution samples that are more informative for generalization and robustness.

To address these inefficiencies, we adopt the Selective Language Modeling (SLM) paradigm (Lin et al., 2024), which optimizes the model over a selected subset of tokens per training step. Formally, let $\pi_\phi(x)$ be a token selection policy that produces a binary mask $m = (m_1, \ldots, m_T) \in \{0,1\}^T$ for an input sequence $x$, the SLM objective becomes:

$$\mathcal{L}_{\text{SLM}}(\theta, \phi) = \mathbb{E}_{x \sim \mathcal{D}, m \sim \pi_\phi(x)} \left[ \sum_{j=1}^T m_j \cdot \ell_\theta(x_j) \right],$$

where $\ell_\theta(x_j)$ is the per-token loss given in (ii). Existing approaches typically rely on learned or reference model ($\phi$)-based policies for $\pi_\phi$, that are expensive to train and may not generalize well (Lin et al., 2024), or design offline selectors (Xie et al., 2023b; Wettig et al., 2024). In contrast, we propose ESLM, a self-supervised online SLM framework rooted in statistical risk that eliminates the need for an auxiliary external selector to improve computational efficiency.

ESLM reshapes the loss towards *high-risk* inputs within each batch by dynamically filtering training signals both at the instance and token levels using risk-based thresholds derived from empirical batch distributions. Concretely, the risk is characterized by either $(i)$ high predictive uncertainty (VaR-entropy selection) or $(ii)$ high loss impact (CVaR-loss selection). At each step, a candidate batch $\mathcal{B} = \{x_1, \ldots, x_M\} \sim \mathcal{D}$ is sampled. A shallow early-exit proxy model $\theta^L$ (first $L$ layers of $\theta$) is used to compute per-token risk scores (entropy (i) or loss (ii)), which are aggregated into per-instance scores: $S_{\theta^L}(x_j) = \frac{1}{T} \sum_{t=1}^{T} S_{\theta^L}(x_j^t)$. Given the empirical score distribution $\hat{\mathbb{P}}_\mathcal{B}$ over the batch, a VaR threshold at confidence level $\alpha$ is then applied to select high-risk instances:

$$\bar{\mathcal{B}} = \{x_j \in \mathcal{B} \mid S_{\theta^L}(x_j) \geq S_{\theta^L,\alpha}^{\mathrm{VaR}}\}, \text{ where } S_{\theta,\alpha}^{\mathrm{VaR}} = \inf\left\{\tau \in \mathbb{R} \;\middle|\; \hat{\mathbb{P}}_\mathcal{B}(S_{\theta^L}(x_j) \geq \tau) \leq 1-\alpha\right\}.$$

On the reduced batch $\bar{\mathcal{B}}$, the model $\theta$ computes per-token risk scores $S_\theta(x_j^t)$ using either entropy (VaR-entropy) or loss (CVaR-loss). Given the empirical score distribution $\hat{\mathbb{P}}_{\bar{\mathcal{B}}}$ over the batch, a VaR threshold is applied at the token level, which defines a high-risk subset $\tilde{\mathcal{B}} = \{x_j^t \in \bar{\mathcal{B}} \mid S_\theta(x_j^t) \geq S_{\theta,\alpha}^{\mathrm{VaR}}\}$, and an associated normalized training distribution: $Q_\tau \in \mathcal{P}_\alpha(\bar{\mathcal{B}}; \theta)$, $Q_\tau(x_j^t) \propto \mathbb{1}[x_j^t \in \tilde{\mathcal{B}}]$, where $\tau$ corresponds to the minimizer defined in (1). The training proceeds by minimizing:

$$\mathcal{L}_{\tilde{\mathcal{B}}}(\theta) = \mathbb{E}_{\bar{\mathcal{B}} \sim \mathcal{D}}\left[\mathbb{E}_{x_j^t \sim Q_\tau}\left[\ell_\theta(x_j^t)\right]\right] = \mathbb{E}[\ell_\theta(x_j^t) \mid x_j^t \in \tilde{\mathcal{B}}],$$

which corresponds to CVaR (in (2)) when risk is based on token-level loss, and to an uncertainty-weighted loss when based on entropy. Our approach is detailed in Algorithm 1.

**ESLM variations.** While both VaR-entropy and CVaR-loss strategies select the upper tail of their respective score distributions; their inductive biases differ. The CVaR-loss selection emphasizes high-loss inputs, including both confidently incorrect predictions and uncertain correct ones. This helps the model correct overconfident mistakes and calibrate uncertainty. In contrast, the VaR-entropy selection focuses purely on predictive uncertainty, regardless of correctness, promoting learning in ambiguous or underexplored regions. We illustrate these differences through qualitative examples in Appendix F, showing that ESLM selects rare, or semantically rich tokens across domains.

The following formulations apply to token-level selection; however, instance-level selection is similar.

**Bilevel game formulation.** ESLM can be framed as a two-player adversarial game between the *model* and a *masker* (*adversary*). This provides a bilevel optimization perspective where the masker perturbs the effective training distribution by choosing a threshold $\tau$ that determines which tokens are selected for training, under the VaR constraint, and the model minimizes its loss over the induced sub-distribution. Formally, the training process can be written as follows:

$$\min_\theta \mathbb{E}_{\bar{\mathcal{B}} \sim \mathcal{D}}\left[\mathbb{E}_{x_j^t \sim Q_\tau}\left[\ell_\theta(x_j^t)\right]\right] \text{ subject to } \tau \in \arg\min_{\tilde{\tau} \in \mathbb{R}}\left\{\tilde{\tau} \;\middle|\; \hat{\mathbb{P}}_{\bar{\mathcal{B}}}\left(S_\theta(x_j^t) \geq \tilde{\tau}\right) \leq 1-\alpha\right\}, \quad (3)$$

where $\hat{\mathbb{P}}_{\bar{\mathcal{B}}}$ is the empirical risk score distribution over the batch. This structure defines adversarial dynamics where the masker restricts the model to optimize over the most challenging subset of tokens, forcing it to improve performance on the tail distribution, and the model adapts to this shift. When the score function is $S_\theta(x_j^t) = \ell_\theta(x_j^t)$ (CVaR-loss), this procedure minimizes the CVaR at level $\alpha$, thereby linking ESLM to classical risk-sensitive learning (Curi et al., 2020; Gagne and Dayan, 2021).

**Distributionally robust optimization interpretation.** ESLM admits a distributionally robust optimization (Duchi and Namkoong, 2021; Kuhn et al., 2025) interpretation. VaR thresholding restricts the training loss to a subset of tokens within the batch—those with scores in the top $(1-\alpha)$ quantile. This induces an adversarial sub-distribution $Q$ over the batch, supported only on the most challenging tokens. ESLM can then be seen as minimizing the worst-case expected loss over this ambiguity set:

$$\min_\theta \sup_{Q \in \mathcal{P}_\alpha(\bar{\mathcal{B}}; \theta)} \mathbb{E}_{x_j^t \sim Q}\left[\ell_\theta(x_j^t)\right]$$

$$\text{where } \mathcal{P}_\alpha(\bar{\mathcal{B}}; \theta) = \left\{Q \ll \hat{\mathbb{P}}_{\bar{\mathcal{B}}} \;\middle|\; \mathrm{supp}(Q) \subseteq \{x_j^t \in \bar{\mathcal{B}} \mid S_\theta(x_j^t) \geq S_{\theta,\alpha}^{\mathrm{VaR}}\}\right\}.$$

This robust optimization perspective explains why ESLM improves generalization: by optimizing performance under adversarial distributions, the model develops robustness to distributional shifts. The experimental results in Section 5.1 also confirm consistent generalization improvements by ESLM.

**Risk-aware metric intuition.** Unlike fixed or heuristic input selection thresholds, ESLM employs a quantile-based cutoff on the batch score distribution, selecting instances and tokens that fall within the high-risk tail. Aside from connecting ESLM to distributionally robust optimization, this design choice offers several advantages over the arbitrary thresholds: $(i)$ the VaR/CVaR formulation provides a statistically principled mechanism to identify informative and difficult tokens, yielding both statistical guarantees on coverage and robustness to distribution shifts, $(ii)$ the tail-risk formulation improves generalization (Section 5.1), a utility that extends beyond pure computational efficiency. Furthermore, by dynamically adjusting risk-awareness through the confidence level $\alpha$, a token-level curriculum that balances computational efficiency and generalization can be derived, which we show with an adaptive extension of ESLM in Section 3.1.

**Implementation and computational cost.** We implement ESLM at the mini-batch level, compatible with distributed training. Each step has two passes: we first run an early-exit proxy forward with the first $L$ transformer blocks of the same model in inference mode. From this shallow pass, we compute per-token statistics and aggregate to per-instance scores (mean over non-padded tokens). We then apply a VaR threshold over the $M$ instance scores to select the subset. In the training pass, we run the full model on the selected instances and apply token-level loss shaping as given in Algorithm 1. The computational overhead from VaR thresholding is minimal, requiring $O(M \log M)$ time per batch (with batch size $M$). This cost is negligible compared to the dominant forward/backward FLOPs of the main model. We discuss the runtime overhead in Appendix D.3.

**FLOPs accounting.** Following Kaplan et al. (2020); Chowdhery et al. (2023), a full forward–backward pass costs $6N + 12LHQT$ FLOPs per token, for an LM with $N$ parameters, $L$ layers, $H$ attention heads, head size $Q$, and sequence length $T$. In the proxy phase, we run a forward-only early-exit of depth $L' \leq L$, costing $2N_{\text{proxy}} + 6L'HQT$ FLOPs per token, where $N_{\text{proxy}}$ is the parameters for embeddings, layer norms and the first $L'$ transformer blocks. Since the risk scores need logits, we add $2DV$ FLOPs per token for the head projection, with embedding dimension $D$ and vocabulary size $V$. During token-level loss shaping, masked tokens skip part of the backward in the final layer components (head + FFN matmuls + final/pre-FFN layer norms), yielding $\approx 4DV + 4D + 32D^2 + 4D$ FLOPs saving *per masked token* (Kaplan et al., 2020). ESLM's optimized FLOPs savings come primarily from $(i)$ dropping whole instances after the proxy pass, which avoids their full forward–backward in the training phase; and $(ii)$ skipping last-block gradient computation for loss-masked tokens.

**Downstream impact.** Effective pretraining increasingly hinges on how data is selected (Tirumala et al., 2023; Mayilvahanan et al., 2025). Improvements in training loss does not guarantee better downstream generalization, particularly under distribution shift (Ramanujan et al., 2023; Isik et al., 2025). ESLM addresses this by providing hierarchical control over which parts of the input receive focus, concentrating optimization to high-risk tokens. In Section 5.1, we demonstrate that ESLM improves both loss–vs–compute efficiency and downstream performance than standard training.

**Token vs instance-level selection.** Unlike methods that filter or reweight entire sequences (Wang et al., 2024; Sow et al., 2025), ESLM uses a *hierarchical* scheme: a cheap proxy pass prunes low-value instances, then loss is shaped at the finer granularity of tokens. This combines compute-awareness with information-awareness, retaining useful tokens within kept sequences. ESLM is natively compatible with autoregressive training and avoids data pipeline changes. Under same compute budget, this hierarchical selection generalizes better than coarse instance-only selection methods (Section 5.1).

## 3.1 ADA-ESLM: ADAPTIVE CONFIDENCE THRESHOLDING

While a fixed confidence level $\alpha$ in ESLM yields strong efficiency gains (see Section 5.1), the optimal $\alpha$ level may vary throughout training. Early in training, broad token coverage may improve generalization, whereas later stages benefit from focusing on harder or more informative tokens. To accommodate this, we introduce ADA-ESLM, a dynamic variant that adjusts $\alpha$ during training using a *risk-sensitive controller* driven by CVaR feedback. In each evaluation step $k$, we compute $\text{CVaR}_{\alpha_k}$ of the per-token risk scores, using (2). We then track the changes in CVaR to detect shifts in training difficulty, estimated from model training dynamics, and update $\alpha$ using a multiplicative rule:

$$\alpha_{k+1} \leftarrow \alpha_k \cdot \exp(-\gamma \cdot \Delta_{\text{norm}}(\alpha_k)), \text{ where } \Delta_{\text{norm}}(\alpha_k) := \frac{\text{CVaR}_{\alpha_k} - \text{CVaR}_{\alpha_{k-1}}}{\text{CVaR}_{\alpha_{k-1}} + \varepsilon}.$$

Here, $\Delta_{\mathrm{norm}}(\alpha_k)$ is a dimension and scale-independent signal capturing the relative change in CVaR, $\gamma > 0$ controls adaptation rate, and $\varepsilon$ is a small constant for numerical stability. The core idea for this update rule is *stabilizing* CVaR: if $\Delta_{\mathrm{norm}} > 0$ (i.e., CVaR increases), the model is encountering harder tokens, $\alpha$ is then decreased to include more tokens and expand the training signal. Conversely, if $\Delta_{\mathrm{norm}} < 0$, the model is improving on difficult tokens. We increase $\alpha$ to focus learning on high-risk tokens. ADA-ESLM extends the adversarial game in (3) by equipping the masker with a CVaR-driven controller that adapts token sparsity in response to training dynamics, offering a form of curriculum learning. We provide the ADA-ESLM algorithm in Appendix B (see Algorithm 2).

### 3.2 ESLM-KD: RISK-AWARE KNOWLEDGE DISTILLATION WITH ESLM

Knowledge distillation transfers knowledge from a teacher model to a student by encouraging the student to match the teacher's output distribution (Buciluǎ et al., 2006; Hinton, 2015). In language modeling, Rawat et al. (2024) showed that a small LM supervision improves the training of a much more capable LLM. While the standard framework operates over all tokens—typically using sequence- or word-level KL divergence (Kim and Rush, 2016)—we can utilize ESLM for risk-aware distillation.

To this end, we provide a use case, ESLM-KD, with the implementation details presented in Algorithm 3 in Appendix C. Specifically, we apply $\mathrm{VaR}_\alpha$ thresholding using a student LM to select high-risk tokens, which are then used to compute the KL divergence between teacher and student logits. The student is trained only on these selected tokens, focusing its capacity on uncertain or error-prone regions. This strategy is teacher-agnostic, relying on the internal statistics of the student model for selection. ESLM-KD generates a sparse supervision signal based on selected tokens, resulting in improved compute and sample efficiency, as further empirically supported in Section 5.1.1.

## 4 RELATED WORK

**Online data subset selection.** Efficient data selection is essential for scaling LLM pretraining, where full-corpus training is often prohibitively expensive (Albalak et al., 2024). While early work focused on static or offline methods, such as filtering (Marion et al., 2023) or scoring examples before training (Coleman et al., 2020; Xie et al., 2023b; Wettig et al., 2024) or during fine-tuning (Xia et al., 2024), such methods lack adaptability and struggle to scale in large-batch or continual pretraining settings. Online data selection overcomes these limitations by adapting to the evolving state of the model. Early strategies on online example-level selection prioritized high-loss samples to accelerate convergence (Loshchilov and Hutter, 2015; Katharopoulos and Fleuret, 2018; Jiang et al., 2019) or leveraged gradients (Killamsetty et al., 2021). Recent works (Mindermann et al., 2022; Wang et al., 2024) apply gradient-based influence scoring (Sachdeva et al., 2024) to guide instance selection or leverage reference models for token selection (Fan and Jaggi, 2023; Lin et al., 2024); however, they often incur high memory due to expensive gradient computations or additional supervision costs from curated reference models and validation sets. In contrast, ESLM introduces a *lightweight*, self-supervised, hierarchical batch selection mechanism, eliminating offline preprocessing, external supervision, or costly gradient tracing. This yields an easily integrable approach that achieves a favorable trade-off between compute efficiency and robustness, while remaining agnostic to training configurations.

**Risk-aversion in language modeling.** Risk-sensitive optimization offers a principled mechanism to enhance robustness by focusing training on high-risk examples (Rockafellar et al., 2000). The CVaR objective has been previously studied in classification (Curi et al., 2020), submodular optimization (Maehara, 2015), and fair learning (Williamson and Menon, 2019), typically to mitigate the influence of tail-risk or worst-case samples. However, in the context of language modeling, CVaR-based approaches remain relatively underexplored. Notable exceptions include methods (Oren et al., 2019) that aggregate losses over topics to address distributional shift but these typically operate at the group level, or in fine-tuning LLMs with reinforcement learning (Chaudhary et al., 2024). On the contrary, ESLM brings risk-aware optimization to the token level for LLM pretraining. Each batch is shaped into a high-risk sub-distribution by the fine-grained risk control of ESLM, incorporating a distributionally robust view of token-level optimization. Unlike heuristic loss-based filtering, ESLM offers a theoretically grounded and practical approach for efficient and robust large-scale pretraining under uncertainty.

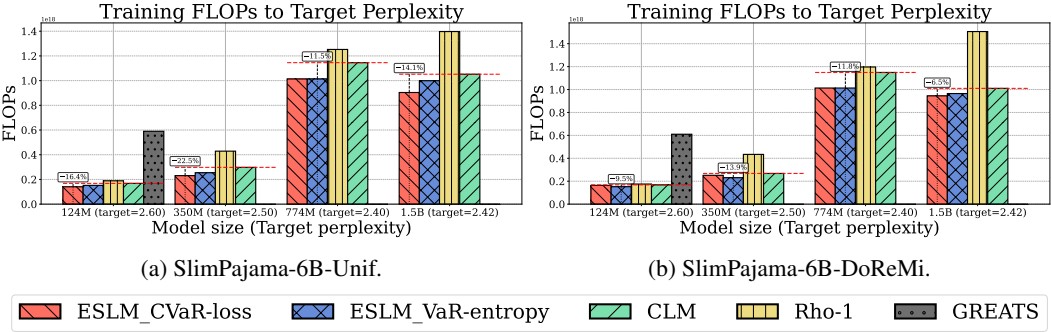

(a) SlimPajama-6B-Unif.  (b) SlimPajama-6B-DoReMi.

Figure 2: **Training FLOPs (↓) required for convergence to target validation (log) perplexity.** We report training FLOPs required by the {124M, 350M, 774M, 1.5B} parameter models to converge to a target validation loss threshold across datasets. The % labels show the percentage FLOPs savings provided by ESLM relative to the best performing baseline. ESLM reduces training cost by focusing optimization on the high-risk instances and tokens and eliminating redundant gradient computation.

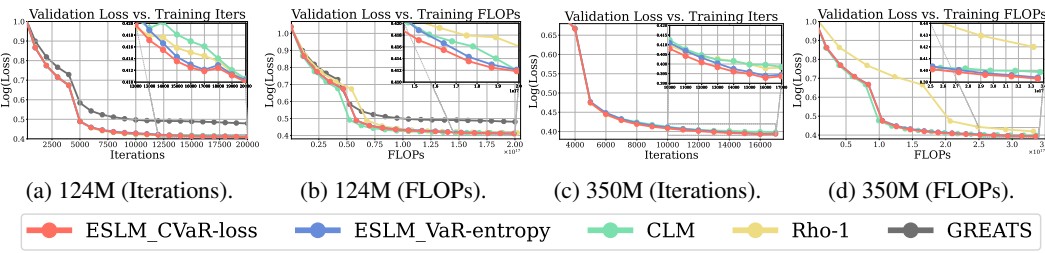

(a) 124M (Iterations).   (b) 124M (FLOPs).   (c) 350M (Iterations).   (d) 350M (FLOPs).

Figure 3: **Validation loss convergence.** We report convergence of validation loss versus training FLOPs/iterations of models trained on SlimPajama-6B-Unif mixture. ESLM variants provide faster convergence to lower loss values in terms of iterations, while requiring fewer FLOPs than baselines at the convergence. See Appendix E.1 for additional results on other pretraining corpora.

## 5 EXPERIMENTS

In this section, we evaluate ESLM with two variants (ESLM-CVaR-loss and ESLM-VaR-entropy) across diverse pretraining settings—varying model scales, data mixtures, and training budgets—to assess its impact on both efficiency and generalization.

In our experiments, we use the SlimPajama-6B (Soboleva et al., 2023) dataset: a 6B token mixture spanning seven domains {Arxiv, Book, CommonCrawl, C4, GitHub, StackExchange, Wikipedia}, used with both uniform and DoReMi (Xie et al., 2023a) domain weights (see Appendix D.2 for the exact weight values).

**Experimental setup.** We pretrain GPT-2 models with 124M, 350M,774M, and 1.5B parameters using a BPE tokenizer (Sennrich et al., 2016) with vocabulary size 50,304. All models are trained with a sequence length of 1024, gradient accumulation over 40 steps, and mini-batch sizes of {4,8,12}. We use AdamW with cosine learning rate decay; full hyperparameters are provided in Appendix D.1. We apply ESLM with confidence level $\alpha$ tuned over the set $\{0.05, 0.1, 0.2\}$. Additional results with varying $\alpha$ levels are presented in Section 5.2.

**Baselines.** We compare ESLM variants against regular training and online batch selection methods: **(1)** CLM, introduced in Section 2, **(2)** Rho-1 (Lin et al., 2024), an online SLM using a reference model to score token loss differentials, **(3)** GREATS (Wang et al., 2024), a state-of-the-art online sample selection method based on high-quality validation data and per-sample gradients. GREATS' high memory requirements, even with ghost inner product optimizations, limited our comparisons to the 124M setting. For distillation experiments (Section 5.1.1), we further compare against the dense distillation and SALT (Rawat et al., 2024) methods. We provide the baseline details in Appendix D.4.

**Performance metrics.** We assess our method concerning training efficiency and generalization ability by tracking the metrics: $(i)$ training FLOPs required to converge to target validation perplexity, $(ii)$ validation loss convergence versus training FLOPs/iterations, and $(iii)$ zero-/few-shot accuracy

Table 1: **Generalization performance on downstream tasks.** All models (124M) are pretrained under a $\sim$3E17 FLOPs budget on SlimPajama-6B-Unif mixture. We report the best observed accuracy$_{(\text{standard error})}$ or exact match if provided, during training. **Highlighted** values indicate the best performance. See Appendix E.2 for the results under various model sizes and datasets.

| Benchmark | | Method (124M) | | | | |
|---|---|---|---|---|---|---|
| | # Shots | ESLM-CVaR-loss | ESLM-VaR-entropy | CLM | Rho-1 | GREATS |
| ARC-E (Clark et al., 2018) | 0-shot | $0.3712_{(0.0099)}$ | $0.3766_{(0.0099)}$ | $0.3644_{(0.0099)}$ | $0.3657_{(0.0099)}$ | $0.3236_{(0.0096)}$ |
| LAMBADA (Paperno et al., 2016) | 5-shot | $0.1595_{(0.0051)}$ | $0.1721_{(0.0053)}$ | $0.1701_{(0.005)}$ | $0.1680_{(0.005)}$ | $0.0254_{(0.002)}$ |
| SciQ (Welbl et al., 2017) | 5-shot | $0.714_{(0.0143)}$ | $0.71_{(0.0144)}$ | $0.6970_{(0.0145)}$ | $0.7000_{(0.0145)}$ | $0.4350_{(0.0157)}$ |
| HellaSwag (Zellers et al., 2019) | 5-shot | $0.2930_{(0.0045)}$ | $0.2964_{(0.0046)}$ | $0.2901_{(0.0045)}$ | $0.2893_{(0.0045)}$ | $0.2621_{(0.0044)}$ |
| TriviaQA (Joshi et al., 2017) | 1-shot | $0.0128_{(0.0001)}$ | $0.0066_{(0.0006)}$ | $0.0078_{(0.0007)}$ | $0.0090_{(0.0007)}$ | $0.0007_{(0.0002)}$ |
| COPA (Wang et al., 2019) | 5-shot | $0.64_{(0.0482)}$ | $0.62_{(0.0488)}$ | $0.62_{(0.0488)}$ | $0.62_{(0.0488)}$ | $0.64_{(0.0482)}$ |
| MultiRC (Wang et al., 2019) | 5-shot | $0.5633_{(0.0071)}$ | $0.5429_{(0.0072)}$ | $0.5338_{(0.0072)}$ | $0.5338_{(0.0072)}$ | $0.5497_{(0.0071)}$ |
| OpenBookQA (Mihaylov et al., 2018) | 5-shot | $0.164_{(0.0166)}$ | $0.17_{(0.0168)}$ | $0.174_{(0.017)}$ | $0.164_{(0.0166)}$ | $0.148_{(0.0159)}$ |
| PiQA (Bisk et al., 2020) | 5-shot | $0.6240_{(0.0113)}$ | $0.6191_{(0.0113)}$ | $0.6099_{(0.0114)}$ | $0.6180_{(0.0113)}$ | $0.5571_{(0.0116)}$ |
| Average ($\uparrow$) | | **0.39353** | 0.39041 | 0.38523 | 0.38531 | 0.32684 |

(normalized, if provided) in downstream benchmark tasks from the `lm-eval-harness` (Gao et al., 2024) suite, spanning QA, reasoning, and generation. To track convergence behavior, we report the learning trajectory using a running average with a window size of 5 evaluation points. We further evaluate performance across model sizes and dataset mixtures. We estimate training FLOPs as explained in Section 3. The details on metrics and experimental setup are provided in Appendix D.

## 5.1 EXPERIMENTAL RESULTS

In this section, we report the performance of ESLM variants against the baseline methods, followed by the results of its implementation as a knowledge distillation mechanism and ablation analyses.

**Validation loss vs training FLOPs/iterations.** As presented in Figure 2, ESLM consistently requires fewer training FLOPs to reach target validation loss across model sizes and datasets. ESLM provides strong efficiency gains at convergence, reaching the level of, e.g., $-22.5\%$ reduced FLOPs relative to standard training in a 350M setting. Notably, these gains over CLM are essentially free, i.e., ESLM operates under the same optimizer, dataset, with no external supervision. Figure 3 further shows that ESLM accelerates learning convergence compared to the baselines, discovering lower validation loss with fewer iterations. While the proxy phase adds a small upfront cost, subsequent instance pruning and token-level loss shaping drive lower losses and net FLOPs savings at convergence across scales compared to CLM. Unlike Rho-1, which queries an external reference model—adding extra compute overhead and requiring

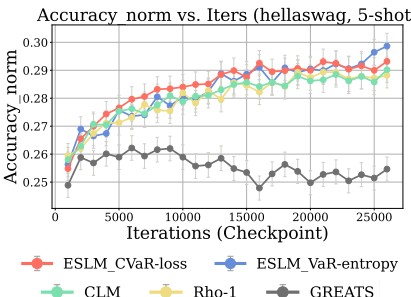

Figure 4: 5-shot accuracy (norm) ($\uparrow$) performance on HellaSwag throughout training. ESLM variants discover higher accuracy levels than baselines, with particular gains in the later training stages.

high-quality data—ESLM instead leverages model-internal training dynamics for selection, avoiding such offline preprocessing. Similarly, while GREATS employs an efficient ghost inner product approximation (Wang et al., 2024), it still relies on curated validation data and per-sample gradient estimates, which are impractical at larger model scales due to high memory demands. In contrast, ESLM operates without gradient tracing and scales naturally. Moreover, GREATS performs selection at the instance level, often discarding informative tokens within partially useful sequences—leading to worse perplexity. This highlights the instance and token-level granularity of ESLM, which avoids this limitation by emphasizing valuable sub-sequence information.

**Downstream performance.** Table 1 summarizes the best zero-/few-shot accuracy achieved by 124M models trained on SlimPajama-6B-Unif under a fixed compute budget of $\sim$3E17 FLOPs. ESLM variants significantly outperform baselines in average accuracy, with consistent gains over GREATS and Rho-1 across all tasks. Figure 4 further illustrates the accuracy norm convergence on the HellaSwag benchmark, where ESLM-CVaR-loss achieves faster early gains, while ESLM-VaR-entropy surpasses baselines in later training stages. These results, including additional evaluations in Appendix E.2, show that ESLM improves both training efficiency and generalization.

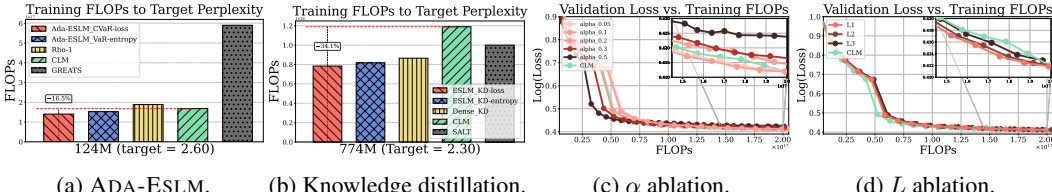

(a) ADA-ESLM.    (b) Knowledge distillation.    (c) $\alpha$ ablation.    (d) $L$ ablation.

Figure 5: **Extended analyses for ESLM. (a):** ADA-ESLM reduces FLOPs for convergence to target validation loss ($\downarrow$) by adaptively tuning $\alpha$ based on training dynamics. **(b):** In knowledge distillation for 774M, ESLM-KD converges to target validation loss with substantially fewer FLOPs. **(c):** Varying the $\alpha$ level enables flexible control over the trade-off between training efficiency and model quality. **(d):** Proxy depth ablation shows that shallow proxies achieve similar convergence performance to deeper alternatives; hence, minimal proxy computation suffices for effective instance selection.

**ADA-ESLM experiments.** We train ADA-ESLM (124M) with $\alpha_0 = 0.1, \gamma = 0.5$ on SlimPajama-6B-Unif. Figure 5a reveals that ADA-ESLM achieves the target validation perplexity with $-16.5\%$ training FLOPs savings at convergence. As detailed in Appendix B (Figure 7), ADA-ESLM provides an implicit curriculum learning: the training process begins with broader input coverage and gradually shifts focus toward higher-risk tokens—without manual scheduling or external supervision. Adaptively adjusting $\alpha$ based on CVaR feedback stabilizes training while offering a principled trade-off between compute-efficiency and generalization. Downstream evaluations in Figure 8 and Table 2 (Appendix B) confirm that ADA-ESLM further achieves higher average downstream accuracy than baselines, improving generalization while maintaining high training efficiency.

### 5.1.1 EXPERIMENTS FOR KNOWLEDGE DISTILLATION WITH ESLM-KD

To utilize ESLM for risk-aware knowledge distillation (Section 3.2), we pretrain a 774M student LM using a 124M teacher on SlimPajama-6B-Unif. We set the distillation weight $\lambda = 0.5$ and teacher temperature $\rho = 1.0$. Training details are provided in Appendix C. We compare ESLM-KD against three baselines: CLM, dense distillation without token selection, and SALT (Rawat et al., 2024), a two-stage distillation-then-pretraining pipeline. As shown in Figure 5b, ESLM-KD models converge to the target validation perplexity with substantially fewer FLOPs; with $-34\%$ FLOPs savings compared to CLM. Furthermore, as reported in Appendix C (Table 3), it outperforms baseline models in downstream tasks, demonstrating the effectiveness of ESLM for efficient and generalizable distillation.

### 5.2 ABLATION AND ADDITIONAL ANALYSES

- **Confidence level ($\alpha$):** In Figure 5c, we assess the sensitivity of ESLM to varying $\alpha$ values: $\{0.05, 0.1, 0.2, 0.3, 0.5\}$. Lower $\alpha$ values improve data coverage but increase computation, whereas higher $\alpha$ levels enhance efficiency at the cost of underutilization. We find $\alpha \in [0.1, 0.2]$ offers a favorable trade-off between compute savings and generalization.

- **Proxy depth ($L$):** In Figure 5d, we assess the effectiveness of ESLM under varying $L$ levels: $\{1, 2, 3\}$ for 124M model. ESLM achieves similar validation loss convergence, hence, proxy computation with $L = 1$ suffices for effective instance selection while minimizing overhead costs.

- **Model size:** Across 124M, 350M, 774M, and 1.5B GPT-2 models, ESLM consistently improves efficiency and generalization (see Figures 2, 3, 4, and Appendix E).

- **Pretraining corpus:** We evaluate ESLM on SlimPajama-6B corpus with different domain mixture weights (uniform and DoReMi). The method generalizes well across corpora (see Figure 2) without requiring domain-specific tuning. Detailed results are reported in Appendix E.

- **Token selection analysis:** To better understand the behavior of ESLM variants, we analyze the selected tokens across different domains in Appendix F, which reveals that ESLM focuses on rare, or contextually ambiguous tokens—supporting its risk-aware design.

- **Instance vs token-level selection:** In Appendix-E.3, we report an ablation decomposing ESLM's FLOPs savings into contributions from proxy instance selection and token-level loss shaping.

- **Robustness to label noise:** To show the broader utility of selective pretraining with high risk inputs, we further examine the robustness properties of ESLM grounded in the DRO framework using label noise injection into the pretraining setup. In practice, real-world pretraining corpora inherently contain noisy tokens from data quality heterogeneity, a challenge in large-scale web data pretraining. For this, we pretrain the models on SlimPajama-6B-Unif with 5% label noise, randomly permuting a portion of the next-token targets. The results in Figure 6 show that even

under noisy supervision, ESLM variants achieve the target validation loss with fewer FLOPs compared to standard training, demonstrating that its risk-aware token selection mitigates the effect of corrupted gradients. This experiment further supports the utility of ESLM's hierarchical risk-aware filtering, not only in improving compute efficiency but also in enhancing robustness to label noise.

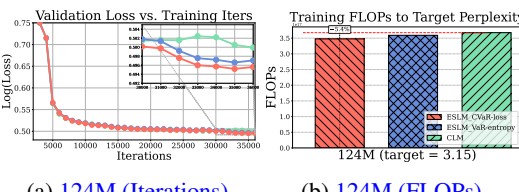

(a) 124M (Iterations).  (b) 124M (FLOPs).

Figure 6: **Pretraining under 5% label noise. (a):** As shown with the learning trajectory using a running average with a window size of five evaluation points, ESLM variants converge faster to lower validation loss values under label noise compared to standard training. **(b):** Even under noisy supervision, ESLM achieves the target validation loss with 5.4% FLOPs savings compared to CLM.

## 6 LIMITATIONS

While ESLM offers an effective approach to selective pretraining, it inherently trades off completeness for efficiency by sparsifying backpropagation. Although this provides compute savings (FLOPs), in practice, it may underutilize the full training signal. As we discuss in Appendix D.3, integrating ESLM with sparsity-aware accelerators could further enhance resource utilization. Consistent with recent efficiency works (Lasby et al., 2024), achieving practical acceleration requires optimized hardware support for sparse backpropagation—an engineering direction beyond the paper's scope.

## 7 CONCLUSION

We introduce ESLM, a selective language modeling with risk-aware hierarchical batch selection, which improves compute efficiency and generalization of LLM pretraining. Rather than training uniformly over all tokens, ESLM applies a risk-sensitive VaR threshold to prioritize high-utility tokens and skip redundant ones during backpropagation. This data-centric strategy effectively improves loss-per-FLOP efficiency, without modifying the model, optimizer, or dataset. By focusing optimization on the most informative inputs, ESLM improves generalization and enhances scalability in language modeling. As a future work, ESLM—along with its ADA-ESLM variant and integration with knowledge distillation—opens new directions in risk-aware token-level curriculum learning, adaptive compute allocation, and risk-aware data valuation for sustainable and efficient LLM scaling.

## ETHICS & SOCIETAL IMPACT STATEMENT

Our work introduces ESLM, a selective language modeling framework which improves training efficiency and generalization performance in LLM pretraining via risk-aware instance and token selection. On the one hand, ESLM enables compute-efficient training by focusing optimization on the most informative parts of the input. This could reduce the energy footprint of large-scale training runs, make LLM development more accessible to institutions with limited compute budgets, and improve model robustness—particularly in out-of-distribution scenarios.

On the other hand, improved training efficiency may accelerate the development of powerful generative models, some of which could be misused for disinformation, synthetic media, or other harmful applications. In addition, token-level filtering methods—if miscalibrated—may reinforce spurious patterns or underrepresent minority language phenomena, inadvertently encoding or amplifying societal biases in the training data. Although ESLM is not tied to a specific application, its performance gains could boost the downstream impact of any application built upon the pretrained models. As our method provides purely training-time improvement, it does not increase model capacity or inference capability directly, which partially limits its risk surface.

Finally, our study does not involve intervention with or collection of data from human participants. Our experiments use widely adopted language-model pretraining corpora or mixtures assembled from publicly available sources under their respective licenses, with provided references.

## REPRODUCIBILITY STATEMENT

For reproducibility, we provide the algorithmic description of ESLM in Algorithm 1 in Section 3. We further explain the implementation details in a separate section for reproducibility in Appendix D.6. In the same section, we provide a reference to the open source code repository on which our implementation is based, the evaluation benchmark suite used for evaluation, and specific implementation mechanisms. Furthermore, in Section 5, we detail the specific experimental setup along with references, specifically the type of model, pretraining corpus and specific dataset mixture weights (see also Appendix D.1-D.2) we use in our experiments.

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

# Appendix

## Table of Contents

## A    STATEMENT FOR THE USE OF LARGE LANGUAGE MODELS (LLMS)

We used LLMs solely as an assistive tool for polishing the writing and improving the readability of this paper. All research ideas, experimental design, implementation, and analysis were conducted by the authors. The paper itself directly studies an online, risk-aware batch selection method for improving the efficiency and distributional robustness of LLM pretraining.

## B    ADA-ESLM: ADAPTIVE CONFIDENCE THRESHOLDING

In Algorithm 2, we provide the algorithmic description of ADA-ESLM. Instead of using a fixed confidence level $\alpha$ throughout training, ADA-ESLM introduces a feedback-driven update mechanism that adjusts $\alpha$ based on the evolving difficulty of the training process. The underlying principle is to achieve a steady state through stabilizing the CVaR signal over time, allowing the model to gradually shift from broad token coverage to a more focused, high-risk subset. When CVaR increases over training intervals, it signals that the model is encountering more difficult (high-risk) examples, requiring a broader coverage. Conversely, a decrease in CVaR suggests that the model is improving on difficult tokens and can afford to focus more narrowly.

Concretely, at each evaluation step $k$ (defined by the interval $T_{\text{eval}}$), ADA-ESLM measures the change in CVaR average tail token-level risk scores, which is a proxy for difficulty. Let $\text{CVaR}_{\alpha_k}$ denote the CVaR value at iteration $k$ computed via (2). We define the normalized CVaR change, $\Delta_{\text{norm}}$, via:

$$\Delta_{\text{norm}}\left(\alpha_k\right) := \frac{\text{CVaR}_{\alpha_k} - \text{CVaR}_{\alpha_{k-1}}}{\text{CVaR}_{\alpha_{k-1}} + \varepsilon},$$

which is a dimension and scale-independent feedback signal, and $\varepsilon > 0$ is a small constant for numerical stability. The controller updates the confidence level $\alpha$ multiplicatively using:

$$\alpha_{k+1} = \alpha_k \cdot \exp\left(-\gamma \cdot \Delta_{\text{norm}}\left(\alpha_k\right)\right),$$

where $\gamma > 0$ controls the update rate. The update rule captures the key intuition:

- If $\Delta_{\text{norm}} > 0$ (CVaR increases), then $\alpha$ is decreased to expand the input selection.
- If $\Delta_{\text{norm}} < 0$ (CVaR decreases), then $\alpha$ is increased to narrow focus to high-risk inputs.

This dynamic adjustment results in a form of *token-level curriculum learning* in which the model begins with broad exposure and progressively narrows focus to the most informative regions of the data. As we further show in Figure 7, ADA-ESLM gradually increases $\alpha$ over training and converges to a stable operating regime in the range $[0.1, 0.2]$—a region empirically shown to yield a strong trade-off between training efficiency and data utility (Section 5.2, Figure 5c).

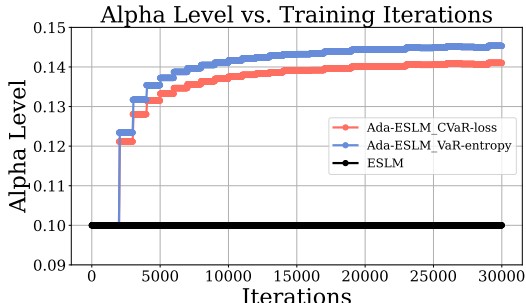

Figure 7: **ADA-ESLM confidence level ($\alpha$) during training.** ADA-ESLM adjusts $\alpha$ dynamically using a CVaR-based controller to stabilize training. The learned $\alpha$ values converge to the $[0.1, 0.2]$ range—previously shown in Section 5.2 (Figure 5c) to balance training efficiency and data utilization.

In Table 2, we present the downstream performance of ADA-ESLM across standard benchmarks. Notably, the adaptive variant **ADA**-ESLM-VaR-entropy consistently outperforms both baseline methods (CLM, Rho-1, GREATS) and fixed-$\alpha$ ESLM variants. These results highlight the benefit of dynamically adjusting the token selection threshold during training, demonstrating that ADA-ESLM improves generalization while maintaining high training efficiency.

---

**Algorithm 2** ADA-ESLM

1: **Input:** Language model $\theta$, dataset $\mathcal{D}$, learning rate $\eta$, initial confidence level $\alpha_0 \in (0, 1)$, proxy $L$ layers, sensitivity $\gamma > 0$, evaluation interval $T_{\text{eval}}$, batch size $M$, small constant $\varepsilon > 0$.

2: Initialize: $\text{CVaR}_0 \leftarrow 0$.

3: Initialize the list: `CVaR_history` $\leftarrow [\,]$.

4: Append $\text{CVaR}_0$ to `CVaR_history`.

5: **for** each training iteration $k = 1, \ldots, K$ **do**

6:     Sample a batch of instances $\mathcal{B} = \{x_1, \ldots, x_M\} \sim \mathcal{D}$ of length $T \{x_j^t\}_{t=1}^T, \forall j \in \{1, \ldots, M\}$.

7:     Compute per-instance scores $S_{\theta_k^L}(x)$ using early-exit $\theta_k^L$ proxy.     */ Entropy or loss

8:     Compute threshold $S_{\theta_k^L,\alpha}^{\text{VaR}} \leftarrow \text{VaR}_\alpha\left(\{S_{\theta_k^L}(x_j)\}_{j=1}^M\right)$ using (1).

9:     $\bar{\mathcal{B}} \leftarrow \{x_j \in \mathcal{B} \mid S_{\theta_k^L}(x_j) \geq S_{\theta_k^L,\alpha}^{\text{VaR}}\}$.     */ High-risk instance selection

10:     Compute per-token statistics $S_{\theta_k}(x^t)$:     */ Entropy or loss

$$S_{\theta_k}(x_j^t) = \begin{cases} H_{\theta_k}(x_j^t) \text{ as in (i),} & \text{(VaR-entropy)} \\ \ell_{\theta_k}(x_j^t) \text{ as in (ii),} & \text{(CVaR-loss)} \end{cases}$$

11:     Compute threshold $S_{\theta_k,\alpha}^{\text{VaR}} \leftarrow \text{VaR}_\alpha\left(\{\{S_{\theta_k}(x_j^t)\}_{j\in\bar{\mathcal{B}}}\}_{t=1}^T\right)$ using (1).

12:     $\tilde{\mathcal{B}} \leftarrow \{x_j^t \in \bar{\mathcal{B}} \mid S_{\theta_k}(x_j^t) \geq S_{\theta_k,\alpha}^{\text{VaR}}\}$.     */ High-risk token selection

13:     Compute loss over selected tokens:     */ Shaped loss

$$\mathcal{L}_{\tilde{\mathcal{B}}}(x; \theta_k) = \begin{cases} \mathbb{E}[\ell_{\theta_k}(x_j^t) \mid x_j^t \in \tilde{\mathcal{B}}], & \text{(VaR-entropy)} \\ \text{CVaR}_\alpha(\ell_{\theta_k}(x)) = \mathbb{E}[\ell_{\theta_k}(x_j^t) \mid x_j^t \in \tilde{\mathcal{B}}] \text{ using (2),} & \text{(CVaR-loss)} \end{cases}$$

14:     Update model parameters using optimizer $O$: $\theta_{k+1} \leftarrow O(\theta_k, \nabla_\theta \mathcal{L}_{\tilde{\mathcal{B}}}(x; \theta_k), \eta)$.

15:     **if** $k \bmod T_{\text{eval}} = 0$ **then**     */ Update $\alpha$

16:         Compute $\text{CVaR}_{\alpha_k} \leftarrow \text{CVaR}_{\alpha_k}(S_{\theta_k})$ using (2).

17:         Retrieve $\text{CVaR}_{\alpha_{k-1}} \leftarrow$ `CVaR_history`$[-1]$.

18:         Compute normalized CVaR change:

$$\Delta_{\text{norm}}(\alpha_k) \leftarrow \frac{\text{CVaR}_{\alpha_k} - \text{CVaR}_{\alpha_{k-1}}}{|\text{CVaR}_{\alpha_{k-1}}| + \varepsilon}$$

19:         Update confidence level: $\alpha_{k+1} \leftarrow \alpha_k \cdot \exp(-\gamma \cdot \Delta_{\text{norm}}(\alpha_k))$.

20:         Append $\text{CVaR}_{\alpha_k}$ to `CVaR_history`.

21:     **end if**

22: **end for**

23: **return** $\theta_K$

---

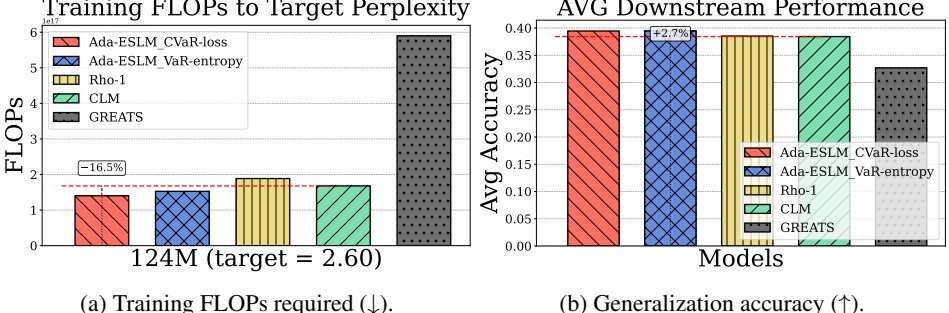

(a) Training FLOPs required ($\downarrow$).         (b) Generalization accuracy ($\uparrow$).

Figure 8: **ADA-ESLM efficiency and generalization performance.** **(a):** ADA-ESLM adaptively tunes the $\alpha$ level based on training dynamics, achieving the target validation (log) perplexity with fewer training FLOPs compared to baselines. **(b):** ADA-ESLM further improves generalization on downstream benchmarks reported in detail in Table 2, achieving higher average accuracy than baselines, trained under the same compute budget.

Table 2: **Generalization performance of ADA-ESLM on downstream tasks.** All models (124M) are pretrained under a $\sim$3E17 FLOPs budget on SlimPajama-6B-Unif mixture. We report the best observed accuracy$_{\text{(standard error)}}$ or exact match if provided, during training. Highlighted values indicate the best performance. Results demonstrate that the dynamic CVaR-driven adjustment of $\alpha$ level leads to improved generalization over baselines, particularly with **ADA**-ESLM-VaR-entropy setting.

| Benchmark | # Shots | **ADA**-ESLM-CVaR-loss | **ADA**-ESLM-VaR-entropy | ESLM-CVaR-loss | ESLM-VaR-entropy | CLM | Rho-1 | GREATS |
|---|---|---|---|---|---|---|---|---|
| | | | | Method (124M) | | | | |
| ARC-E (Clark et al., 2018) | 0-shot | $0.3728_{(0.0099)}$ | $0.3884_{(0.01)}$ | $0.3712_{(0.0099)}$ | $0.3766_{(0.0099)}$ | $0.3644_{(0.0099)}$ | $0.3657_{(0.0099)}$ | $0.3236_{(0.0096)}$ |
| LAMBADA (Paperno et al., 2016) | 5-shot | $0.1631_{(0.0051)}$ | $0.1628_{(0.0051)}$ | $0.1595_{(0.0051)}$ | $0.1721_{(0.0053)}$ | $0.1701_{(0.0051)}$ | $0.1680_{(0.005)}$ | $0.0254_{(0.002)}$ |
| SciQ (Welbl et al., 2017) | 5-shot | $0.6960_{(0.0146)}$ | $0.7060_{(0.0144)}$ | $0.7140_{(0.0143)}$ | $0.7100_{(0.0144)}$ | $0.6970_{(0.0145)}$ | $0.7000_{(0.0145)}$ | $0.4350_{(0.0157)}$ |
| HellaSwag (Zellers et al., 2019) | 5-shot | $0.2976_{(0.0046)}$ | $0.2952_{(0.0046)}$ | $0.2930_{(0.0045)}$ | $0.2964_{(0.0046)}$ | $0.2901_{(0.0045)}$ | $0.2893_{(0.0045)}$ | $0.2621_{(0.0044)}$ |
| TriviaQA (Joshi et al., 2017) | 1-shot | $0.0082_{(0.0007)}$ | $0.0128_{(0.0008)}$ | $0.0128_{(0.0001)}$ | $0.0066_{(0.0006)}$ | $0.0078_{(0.0007)}$ | $0.0090_{(0.0007)}$ | $0.0007_{(0.0002)}$ |
| COPA (Wang et al., 2019) | 5-shot | $0.6600_{(0.0476)}$ | $0.6300_{(0.0485)}$ | $0.6400_{(0.0482)}$ | $0.6200_{(0.0488)}$ | $0.6200_{(0.0488)}$ | $0.6200_{(0.0488)}$ | $0.6400_{(0.0482)}$ |
| MultiRC (Wang et al., 2019) | 5-shot | $0.5600_{(0.0071)}$ | $0.5668_{(0.0071)}$ | $0.5633_{(0.0071)}$ | $0.5429_{(0.0071)}$ | $0.5338_{(0.0072)}$ | $0.5338_{(0.0072)}$ | $0.5497_{(0.0071)}$ |
| OpenBookQA (Mihaylov et al., 2018) | 5-shot | $0.174_{(0.017)}$ | $0.178_{(0.0171)}$ | $0.164_{(0.0166)}$ | $0.170_{(0.0168)}$ | $0.166_{(0.0167)}$ | $0.164_{(0.0166)}$ | $0.148_{(0.0159)}$ |
| PiQA (Bisk et al., 2020) | 5-shot | $0.6191_{(0.0113)}$ | $0.6115_{(0.0114)}$ | $0.6240_{(0.0113)}$ | $0.6191_{(0.0113)}$ | $0.6099_{(0.0114)}$ | $0.6180_{(0.0113)}$ | $0.5571_{(0.0116)}$ |
| **Average ($\uparrow$)** | | **0.39453** | **0.39461** | 0.39353 | 0.39041 | 0.38434 | 0.38531 | 0.32684 |

## C  RISK-AWARE KNOWLEDGE DISTILLATION WITH ESLM-KD

We provide the implementation for our knowledge distillation setup, namely ESLM-KD, in Algorithm 3. The student model $\theta$ computes per-token risk scores over each batch, and high-risk tokens are selected via $\text{VaR}_\alpha$ thresholding. The student is then supervised only on these informative tokens using a combined loss: a weighted sum of KL divergence to the teacher ($\phi$) and standard cross-entropy.

In our experiments (Section 5.1.1), we used a 124M GPT-2 model pretrained with the CLM objective (checkpoint 40,000) as the teacher to train a 774M student models on the SlimPajama-6B-Unif dataset. Based on hyperparameter tuning, we set the distillation weight to $\lambda = 0.5$ and the teacher temperature to $\rho = 1.0$. We compare ESLM-KD against three 774M baselines with the same teacher model: standard CLM training, Dense-KD (dense knowledge distillation without token selection), and SALT (Rawat et al., 2024), a two-staged distillation method which employs distillation in the first stage and then transitions to standard pretraining. For the SALT baseline, we set the distillation iterations to 12,000. We trained all distillation-based models with a compute budget of 1E18 FLOPs.

The experimental results in Figure 5b (Section 5.1) show that ESLM-KD achieves the target validation loss with significantly less training FLOPs, demonstrating its efficiency and effectiveness in large-scale distillation. Table 3 further compares the generalization performance of ESLM-KD against dense distillation using the same teacher model. The results show that integrating risk-aware token selection into distillation not only reduces compute cost but also improves downstream accuracy over full-token distillation.

---

**Algorithm 3** ESLM-KD

---

1: **Input:** Teacher LM parameters $\phi$, student LM parameters $\theta$, dataset $\mathcal{D}$, learning rate $\eta$, confidence level $\alpha \in (0,1)$, batch size $M$, teacher temperature $\rho > 0$, distillation loss weight $\lambda \in [0,1]$.
2: **for** each training iteration $k = 1, \ldots, K$ **do**
3:     Sample a batch of tokens $\mathcal{B} = \{x_1, \ldots, x_M\} \sim \mathcal{D}$.
4:     Compute per-token statistics $S_\theta(x_j)$ using the student model:     */ Entropy or loss

$$S_\theta(x_j) = \begin{cases} H_\theta(x_j) \text{ as in (i),} & \text{(VaR-entropy)} \\ \ell_\theta(x_j) \text{ as in (ii),} & \text{(CVaR-loss)} \end{cases}$$

5:     Compute VaR threshold: $S_{\theta,\alpha}^{\text{VaR}} \leftarrow \text{VaR}_\alpha\left(\{S_\theta(x_j)\}_{j=1}^M\right)$ using (1).
6:     Select high-risk tokens: $\tilde{\mathcal{B}} \leftarrow \{x_j \in \mathcal{B} \mid S_\theta(x_j) \geq S_{\theta,\alpha}^{\text{VaR}}\}$.
7:     Compute combined student loss on selected tokens:     */ Distillation + cross-entropy loss

$$\mathcal{L}_{\text{ESLM-KD}} = \frac{1}{|\tilde{\mathcal{B}}|} \sum_{x_j \in \tilde{\mathcal{B}}} \left[\lambda \cdot \text{KL}\left(P_\rho^\phi(x_j \mid x_{<j}) \,\|\, P_\rho^\theta(x_j \mid x_{<j})\right) + (1-\lambda) \cdot \ell_{\theta_k}(x_j)\right]$$

8:     Update student parameters using optimizer $O$: $\theta_{k+1} \leftarrow O(\theta_k, \nabla_\theta \mathcal{L}_{\text{ESLM-KD}}, \eta)$.
9: **end for**
10: **return** $\theta_K$.

---

## D  EXPERIMENT DETAILS

### D.1  EXPERIMENTAL SETUP

We set the training hyperparameters as in Table 4. We train GPT-2 models (Radford et al., 2019) of sizes 124M, 350M, 774M, and 1.5B parameters, with architecture details reported in Table 5.

### D.2  PRETRAINING CORPUS

We utilize the SlimPajama-6B dataset for our experiments. SlimPajama-6B (Soboleva et al., 2023) mixture consists of seven data domains: {Arxiv, Book, CommonCrawl, C4, Github, Stackexchange, Wikipedia} with two weighted versions: uniform domain weights (SlimPajama-6B-Unif) and DoReMi (Xie et al., 2023a) domain weights (SlimPajama-6B-DoReMi).

Table 3: **Generalization performance of ESLM-KD on downstream tasks.** All models (774M) are pretrained under a $\sim$1E18 FLOPs budget on SlimPajama-6B-Unif mixture, using the same teacher model. We report the best observed accuracy$_{\text{(standard error)}}$ or exact match if provided, during training. ==Highlighted== values indicate the best performance.

| Benchmark | # Shots | ESLM-KD-CVaR-loss | ESLM-KD-VaR-entropy | Dense-KD | SALT |
|---|---|---|---|---|---|
| | | | Method (774M) | | |
| ARC-E (Clark et al., 2018) | 0-shot | $0.3901_{(0.01)}$ | $0.3935_{(0.01)}$ | $0.3909_{(0.01)}$ | $0.3947_{(0.01)}$ |
| LAMBADA (Paperno et al., 2016) | 5-shot | $0.2472_{(0.006)}$ | $0.2480_{(0.006)}$ | $0.2429_{(0.006)}$ | $0.2258_{(0.0058)}$ |
| SciQ (Welbl et al., 2017) | 5-shot | $0.766_{(0.0134)}$ | $0.773_{(0.0133)}$ | $0.759_{(0.0135)}$ | $0.770_{(0.0133)}$ |
| HellaSwag (Zellers et al., 2019) | 5-shot | $0.3200_{(0.0047)}$ | $0.3189_{(0.0047)}$ | $0.3179_{(0.0046)}$ | $0.3313_{(0.0047)}$ |
| TriviaQA (Joshi et al., 2017) | 1-shot | $0.0273_{(0.0012)}$ | $0.0280_{(0.0012)}$ | $0.0231_{(0.0011)}$ | $0.0299_{(0.0013)}$ |
| COPA (Wang et al., 2019) | 5-shot | $0.68_{(0.0469)}$ | $0.66_{(0.0476)}$ | $0.69_{(0.0465)}$ | $0.67_{(0.0473)}$ |
| MultiRC (Wang et al., 2019) | 5-shot | $0.5408_{(0.0072)}$ | $0.5406_{(0.0072)}$ | $0.5360_{(0.0072)}$ | $0.5420_{(0.0072)}$ |
| OpenBookQA (Mihaylov et al., 2018) | 5-shot | $0.270_{(0.0199)}$ | $0.290_{(0.0203)}$ | $0.278_{(0.0201)}$ | $0.278_{(0.02)}$ |
| PiQA (Bisk et al., 2020) | 5-shot | $0.6300_{(0.0113)}$ | $0.6245_{(0.0113)}$ | $0.6327_{(0.0112)}$ | $0.6322_{(0.0113)}$ |
| **Average** (↑) | | 0.4301 | **0.4307** | 0.4299 | **0.4304** |

Table 4: Training and evaluation hyperparameters used in all experiments.

| Hyperparameter | Value |
|---|---|
| | **ESLM-Specific** |
| Confidence level ($\alpha$) | Optimized over {0.05, 0.1, 0.2} |
| Early-exit layers for proxy phase ($L$) | Optimized over {1, 2, 3} |
| | **General Setup** |
| Mini-batch size ($M$) in tokens | { 12 (124M), 8 (350M/774M), 4 (1.5B) } $\times$ 1024 |
| Gradient accumulation steps | 40 |
| Effective batch size in tokens | { 480 (124M), 320 (350M/774M), 160 (1.5B) } $\times$ 1024 |
| Sequence length ($T$) | 1024 |
| Vocabulary size ($|\mathcal{V}|$) | 50304 |
| Dropout | 0 |
| Evaluation interval ($T_{\text{eval}}$) | 1000 iterations |
| Evaluation steps | 200 iterations |
| | **Optimization** |
| Optimizer ($O$) | AdamW with $\beta_1 = 0.9$, $\beta_2 = 0.95$ |
| Learning rate schedule | Cosine annealing with warmup |
| Max. learning rate ($\eta$) | 0.0006 (124M/350M), 0.0001 (774M/1.5B) |
| Min. learning rate | 0.00006 |
| Warmup steps | 2000 |
| Decay iterations | 200,000 |
| Weight decay | 0.1 |
| Gradient clipping | 1.0 |
| | **Knowledge Distillation** |
| Teacher temperature ($\rho$) | 1.0 |
| Distillation loss weight ($\lambda$) | 0.5 |

In Table 6, we report the domain weights for the experiments under SlimPajama-6B (Soboleva et al., 2023) mixture, using DoReMi (Xie et al., 2023a) and uniform weights.

### D.3 HARDWARE & COMPUTATIONAL OVERHEAD

All experiments were conducted on the HTCondor-managed cluster equipped with NVIDIA A100 GPUs (80GB). Model pretraining and evaluation were parallelized using PyTorch's Distributed Data Parallel (DDP) framework (Paszke et al., 2019) with the NCCL backend and mixed-precision (bfloat16) training. We used $4 \times$A100 GPUs for experiments on the SlimPajama-6B mixtures.

Table 5: Architecture hyperparameters for GPT-2 model sizes.

| Model size | Layers | Attention heads | Embed dimension |
|---|---|---|---|
| 124M | 12 | 12 | 768 |
| 350M | 24 | 16 | 1024 |
| 774M | 36 | 20 | 1280 |
| 1.5B | 48 | 25 | 1600 |

Table 6: Domain weights used for experiments on the SlimPajama-6B mixture.

| Domain | DoReMi | Unif |
|---|---|---|
| Arxiv | 0.04235 | 0.1428 |
| Book | 0.08201 | 0.1428 |
| CC | 0.381 | 0.1428 |
| C4 | 0.1141 | 0.1428 |
| Github | 0.0654 | 0.1428 |
| Stackexchange | 0.0847 | 0.1428 |
| Wikipedia | 0.2305 | 0.1428 |

**Runtime analysis.** In Table 7, we compare the wall-clock time of 124M models trained on the SlimPajama-6B-Unif, for convergence to a target validation loss value of 2.60. While ESLM achieves substantial reductions in training FLOPs, reaches lower validation loss, and stronger downstream performance, it incurs higher wall-clock time compared to standard training. However, it remains significantly more efficient than GREATS and Rho-1 baselines–nearly twice as fast as Rho-1 and over 8× faster than GREATS. We attribute this overhead to mismatches between sparse training operations and current hardware optimizations.

As also explained by the recent efficiency works (Lasby et al., 2024), achieving practical acceleration from FLOPs savings requires optimized hardware

Table 7: Runtime comparison of 124M models trained on SlimPajama-6B-Unif, for convergence to a target validation loss value of 2.60. The overhead compared to the standard training is mainly due to the mismatch between sparsity introduced via instance & token selection and current hardware optimizations.

| Method | Wall-clock time (hrs) |
|---|---|
| ESLM-VaR-entropy | 11.73 |
| ESLM-CVaR-loss | 9.91 |
| CLM | 5.33 |
| Rho-1 | 17.91 |
| GREATS | 99.89 |

support for sparse backpropagation—an engineering direction which is beyond our paper's scope. Although ESLM's per-token risk scores are computed during the forward pass via sorting with $O(M \log M)$ complexity per batch—without requiring additional external inference or backpropagation—its proxy pass requires a shallow forward call, and VaR-based token filtering introduces sparsity into the training process. This sparsity, while beneficial for compute efficiency, leads to irregular and fragmented backpropagation paths that underutilize the dense compute capabilities of modern accelerators. Unlike the uniform operations of standard CLM, ESLM's selective masking disrupts efficient tensor fusion, resulting in slower wall-clock runtime despite using fewer FLOPs. Nonetheless, ESLM provides a favorable trade-off: improved efficiency per FLOP and enhanced generalization. **We expect future work leveraging sparsity-aware hardware or sparse accelerators to further reduce this overhead (Lasby et al., 2024) and unlock the full potential of selective training.**.

### D.4 BASELINES

We identify the baseline methods against which we compare our ESLM approach, specifically from online batch selection methods for LLM pretraining and standard training as discussed in Section 5. We provide the baseline implementation details below:

- For the Rho-1 baseline (Lin et al., 2024), we used pretrained GPT-2 models trained via CLM objective as the reference model. Since training a high-quality reference model is the main bottleneck of the Rho-1 method, we used the last checkpoints of pretrained models as proxy

models. For pretraining on SlimPajama-6B mixtures (Unif and DoReMi), we used the last saved checkpoint of CLM GPT-2 models as the reference models. Specifically, we utilized 40000, 30000, and 30000 checkpoints for 124M, 350M, 774M models, respectively. We set the loss threshold parameter to $0.1$. For Rho-1's total FLOPs calculation, we include additional FLOPs from the forward call on the reference model.

- For the GREATS baseline, we follow the original setup by Wang et al. (2024), using a small validation set ($0.5\times$ the batch size) and setting the batch selection budget to $0.9$, aligning with ESLM's $\alpha = 0.1$ level. To compute the training FLOPs for GREATS, we include the forward passes on both training and validation inputs, the backward pass through linear layers to obtain per-example gradients with respect to pre-activation outputs, and the additional FLOPs for computing ghost inner products. While GREATS is evaluated only on the GPT-2 124M model in the original paper, we also restrict our comparison to this setting. Despite adopting their ghost inner product optimization, we found the method to be highly memory-intensive when scaling to larger models, and it could not run stably beyond 124M size.

### D.5 EVALUATION DETAILS

We evaluate pretrained models on a suite of standard language understanding benchmarks in the zero-shot and few-shot settings, using the `lm-evaluation-harness` evaluation suite (Gao et al., 2024), including HellaSwag (Zellers et al., 2019), LAMBADA (Paperno et al., 2016), ARC-Easy (Clark et al., 2018), TriviaQA (Joshi et al., 2017), SciQ (Welbl et al., 2017), COPA (Wang et al., 2019), MultiRC (Wang et al., 2019), OpenBookQA (Mihaylov et al., 2018), and PiQA (Bisk et al., 2020) tasks. We used the default settings provided by `lm-evaluation-harness`, which means all evaluations are performed on held-out validation splits, or test splits if provided, and standard errors are calculated using bootstrapping. Accuracy (norm if provided) or exact match is used as the primary metric.

### D.6 REPRODUCIBILITY

For reproducibility, we provide the algorithmic description of the ESLM method in Algorithm 1. Our implementation builds on the open-source `NanoGPT` codebase (Karpathy, 2022). To handle training on the SlimPajama-6B dataset mixture, we adapted the open-source code of DoReMi (Xie et al., 2023a) and DoGE (Fan et al., 2023). As we detail in Section 3, we estimate the training FLOPs based on the theoretical estimate by Kaplan et al. (2020); Chowdhery et al. (2023). To skip redundant gradient computation as a result of token-level loss shaping, we use PyTorch's backward hook mechanism (`Tensor.register_hook(hook)`) for gradient masking. For downstream evaluation, we utilize the publicly available `lm-evaluation-harness` suite (Gao et al., 2024). Our open-source implementation, along with references to the adapted codebases, will be released in the camera-ready version.

## E ADDITIONAL EXPERIMENTAL RESULTS

In this section, we report additional experimental results, showing validation perplexity convergence in compute space (Appendix E.1) and generalization performance in downstream benchmark tasks (Appendix E.2) on different datasets across model sizes.

### E.1 VALIDATION LOSS VERSUS TRAINING FLOPS/ITERATIONS RESULTS

As shown in Figures 9-10, ESLM variants consistently accelerate validation loss convergence in the compute space, requiring fewer training FLOPs to achieve comparable or superior validation loss relative to baseline models. This efficiency gain holds across diverse pretraining corpora and model scales, highlighting the robustness of ESLM across settings.

### E.2 DOWNSTREAM PERFORMANCE EVALUATION RESULTS

Tables 8–12 present the generalization performance of ESLM models ranging from 124M to 774M parameters, trained on different mixtures and evaluated against baseline models on downstream

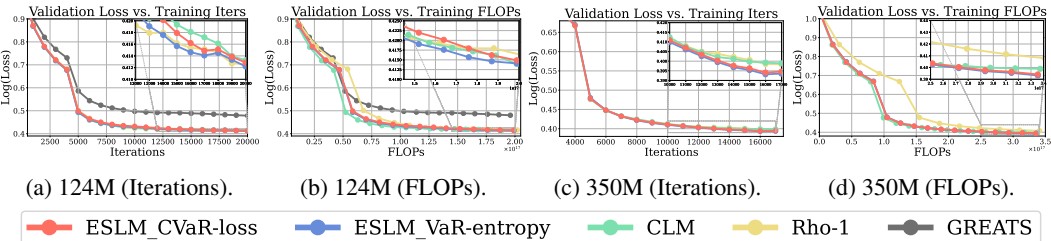

(a) 124M (Iterations).    (b) 124M (FLOPs).    (c) 350M (Iterations).    (d) 350M (FLOPs).

Figure 9: **Validation loss convergence on SlimPajama-6B-DoReMi.** We report convergence of validation loss versus training FLOPs/iterations of models trained on SlimPajama-6B-DoReMi mixture. ESLM variants provide faster convergence to lower loss values in terms of iterations, while requiring fewer FLOPs than baselines at the convergence.

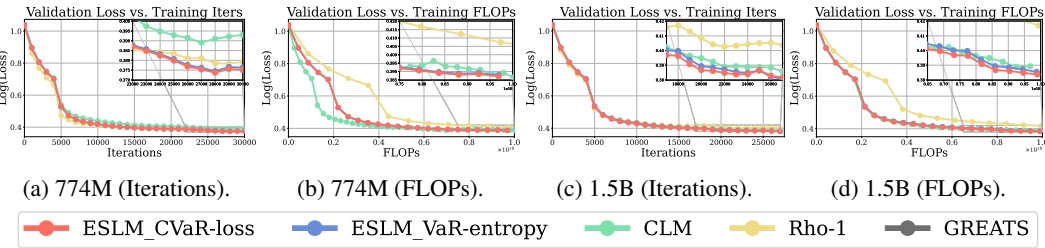

(a) 774M (Iterations).    (b) 774M (FLOPs).    (c) 1.5B (Iterations).    (d) 1.5B (FLOPs).

Figure 10: **Validation loss convergence on SlimPajama-6B-Unif.** We report convergence of validation loss versus training FLOPs/iterations of models trained on SlimPajama-6B-Unif mixture, for 774M and 1.5B models. ESLM variants provide faster convergence to lower loss values in terms of iterations, while requiring fewer FLOPs than baselines at the convergence.

benchmarks. All models are trained under a fixed compute budget measured in training FLOPs. In nearly all settings, ESLM variants consistently outperform baselines, achieving higher average downstream accuracy. While domain mixture weights influence absolute performance, ESLM maintains a consistent advantage, demonstrating that it is not only an efficient and simple approach but also yields better generalization quality.

Table 8: **Generalization performance of 124M models trained on SlimPajama-6B-DoReMi.** All models are pretrained under a $\sim$3E17 FLOPs budget. We report the best observed accuracy$_{(\text{standard error})}$ or exact match if provided, during training. **Highlighted** values indicate the best performance.

| Benchmark | | Method (124M) | | | | |
| --- | --- | --- | --- | --- | --- | --- |
| | # Shots | ESLM-CVaR-loss | ESLM-VaR-entropy | CLM | Rho-1 | GREATS |
| ARC-E (Clark et al., 2018) | 0-shot | $0.3876_{(0.01)}$ | $0.3926_{(0.01)}$ | $0.3838_{(0.01)}$ | $0.3733_{(0.0099)}$ | $0.3190_{(0.0096)}$ |
| LAMBADA (Paperno et al., 2016) | 5-shot | $0.1738_{(0.0053)}$ | $0.1727_{(0.0053)}$ | $0.1581_{(0.0051)}$ | $0.1672_{(0.0052)}$ | $0.0362_{(0.0026)}$ |
| SciQ (Welbl et al., 2017) | 5-shot | $0.731_{(0.014)}$ | $0.718_{(0.0142)}$ | $0.735_{(0.014)}$ | $0.723_{(0.0142)}$ | $0.465_{(0.0158)}$ |
| HellaSwag (Zellers et al., 2019) | 5-shot | $0.2936_{(0.0045)}$ | $0.2924_{(0.0045)}$ | $0.2945_{(0.0045)}$ | $0.2905_{(0.0045)}$ | $0.2639_{(0.0044)}$ |
| TriviaQA (Joshi et al., 2017) | 1-shot | $0.0184_{(0.001)}$ | $0.0145_{(0.0009)}$ | $0.0100_{(0.0007)}$ | $0.0112_{(0.0008)}$ | $0.0007_{(0.0002)}$ |
| COPA (Wang et al., 2019) | 5-shot | $0.65_{(0.0479)}$ | $0.66_{(0.0476)}$ | $0.66_{(0.0476)}$ | $0.64_{(0.0482)}$ | $0.65_{(0.0479)}$ |
| MultiRC (Wang et al., 2019) | 5-shot | $0.5455_{(0.0072)}$ | $0.5367_{(0.0072)}$ | $0.5449_{(0.0072)}$ | $0.5486_{(0.0071)}$ | $0.5309_{(0.0072)}$ |
| OpenBookQA (Mihaylov et al., 2018) | 5-shot | $0.176_{(0.017)}$ | $0.164_{(0.0166)}$ | $0.174_{(0.017)}$ | $0.164_{(0.0166)}$ | $0.148_{(0.0159)}$ |
| PiQA (Bisk et al., 2020) | 5-shot | $0.6169_{(0.0113)}$ | $0.6175_{(0.0113)}$ | $0.6017_{(0.0114)}$ | $0.6033_{(0.0114)}$ | $0.5489_{(0.0116)}$ |
| **Average ($\uparrow$)** | | **0.3992** | **0.3964** | 0.3957 | 0.3912 | 0.3291 |

Table 9: **Generalization performance of 350M models trained on SlimPajama-6B-Unif.** All models are pretrained under a $\sim$3.5E17 FLOPs budget. We report the best observed accuracy$_{(\text{standard error})}$ or exact match if provided, during training. **Highlighted** values indicate the best performance.

| Benchmark | | Method (350M) | | | |
| --- | --- | --- | --- | --- | --- |
| | # Shots | ESLM-CVaR-loss | ESLM-VaR-entropy | CLM | Rho-1 |
| ARC-E (Clark et al., 2018) | 0-shot | $0.4078_{(0.0101)}$ | $0.3973_{(0.01)}$ | $0.4023_{(0.0101)}$ | $0.4006_{(0.0101)}$ |
| LAMBADA (Paperno et al., 2016) | 5-shot | $0.2289_{(0.0059)}$ | $0.2070_{(0.0056)}$ | $0.2095_{(0.0057)}$ | $0.1993_{(0.0056)}$ |
| SciQ (Welbl et al., 2017) | 5-shot | $0.749_{(0.0137)}$ | $0.768_{(0.0134)}$ | $0.744_{(0.0138)}$ | $0.760_{(0.0135)}$ |
| HellaSwag (Zellers et al., 2019) | 5-shot | $0.3301_{(0.0047)}$ | $0.3298_{(0.0047)}$ | $0.3242_{(0.0047)}$ | $0.3207_{(0.0047)}$ |
| TriviaQA (Joshi et al., 2017) | 1-shot | $0.0338_{(0.0013)}$ | $0.0271_{(0.0012)}$ | $0.0329_{(0.0013)}$ | $0.0263_{(0.0012)}$ |
| COPA (Wang et al., 2019) | 5-shot | $0.68_{(0.0469)}$ | $0.68_{(0.0469)}$ | $0.68_{(0.0469)}$ | $0.67_{(0.0473)}$ |
| MultiRC (Wang et al., 2019) | 5-shot | $0.5482_{(0.0071)}$ | $0.5556_{(0.0071)}$ | $0.5492_{(0.0071)}$ | $0.5676_{(0.0071)}$ |
| OpenBookQA (Mihaylov et al., 2018) | 5-shot | $0.276_{(0.02)}$ | $0.288_{(0.0203)}$ | $0.280_{(0.0201)}$ | $0.284_{(0.0202)}$ |
| PiQA (Bisk et al., 2020) | 5-shot | $0.6398_{(0.0112)}$ | $0.6420_{(0.0112)}$ | $0.6349_{(0.0112)}$ | $0.6322_{(0.0113)}$ |
| **Average ($\uparrow$)** | | **0.4326** | **0.4327** | 0.4284 | 0.4289 |

Table 10: **Generalization performance of 350M models trained on SlimPajama-6B-DoReMi.** All models are pretrained under a $\sim$3.5E17 FLOPs budget. We report the best observed accuracy$_{(\text{standard error})}$ or exact match if provided, during training. **Highlighted** values indicate the best performance.

| Benchmark | | Method (350M) | | | |
| --- | --- | --- | --- | --- | --- |
| | # Shots | ESLM-CVaR-loss | ESLM-VaR-entropy | CLM | Rho-1 |
| ARC-E (Clark et al., 2018) | 0-shot | $0.4170_{(0.0101)}$ | $0.4048_{(0.0101)}$ | $0.4196_{(0.0101)}$ | $0.4090_{(0.0101)}$ |
| LAMBADA (Paperno et al., 2016) | 5-shot | $0.2400_{(0.006)}$ | $0.2344_{(0.0059)}$ | $0.2361_{(0.0059)}$ | $0.2144_{(0.0057)}$ |
| SciQ (Welbl et al., 2017) | 5-shot | $0.775_{(0.0132)}$ | $0.794_{(0.0128)}$ | $0.782_{(0.0131)}$ | $0.773_{(0.0133)}$ |
| HellaSwag (Zellers et al., 2019) | 5-shot | $0.3292_{(0.0047)}$ | $0.3271_{(0.0047)}$ | $0.3304_{(0.0047)}$ | $0.3213_{(0.0047)}$ |
| TriviaQA (Joshi et al., 2017) | 1-shot | $0.0357_{(0.0014)}$ | $0.0431_{(0.0015)}$ | $0.0414_{(0.0015)}$ | $0.0395_{(0.0015)}$ |
| COPA (Wang et al., 2019) | 5-shot | $0.68_{(0.0469)}$ | $0.70_{(0.0461)}$ | $0.69_{(0.0465)}$ | $0.68_{(0.0469)}$ |
| MultiRC (Wang et al., 2019) | 5-shot | $0.5457_{(0.0072)}$ | $0.5645_{(0.0071)}$ | $0.5548_{(0.0071)}$ | $0.5558_{(0.0071)}$ |
| OpenBookQA (Mihaylov et al., 2018) | 5-shot | $0.286_{(0.0202)}$ | $0.284_{(0.0202)}$ | $0.280_{(0.0201)}$ | $0.286_{(0.0202)}$ |
| PiQA (Bisk et al., 2020) | 5-shot | $0.6289_{(0.0113)}$ | $0.6414_{(0.0112)}$ | $0.6354_{(0.0112)}$ | $0.6354_{(0.0112)}$ |
| **Average ($\uparrow$)** | | 0.4375 | **0.4437** | 0.4410 | 0.4349 |

Table 11: **Generalization performance of 774M models trained on SlimPajama-6B-Unif.** All models are pretrained under a $\sim$1E18 FLOPs budget. We report the best observed accuracy$_{(\text{standard error})}$ or exact match if provided, during training. **Highlighted** values indicate the best performance.

| Benchmark | Method (774M) | | | | |
|---|---|---|---|---|---|
| | # Shots | ESLM-CVaR-loss | ESLM-VaR-entropy | CLM | Rho-1 |
| ARC-E (Clark et al., 2018) | 0-shot | $0.4040_{(0.0101)}$ | $0.4061_{(0.0101)}$ | $0.4082_{(0.0101)}$ | $0.3985_{(0.01)}$ |
| LAMBADA (Paperno et al., 2016) | 5-shot | $0.2408_{(0.006)}$ | $0.2410_{(0.006)}$ | $0.2336_{(0.0059)}$ | $0.2404_{(0.006)}$ |
| SciQ (Welbl et al., 2017) | 5-shot | $0.763_{(0.0135)}$ | $0.760_{(0.0135)}$ | $0.762_{(0.0135)}$ | $0.780_{(0.0131)}$ |
| HellaSwag (Zellers et al., 2019) | 5-shot | $0.3348_{(0.0047)}$ | $0.3399_{(0.0047)}$ | $0.3333_{(0.0047)}$ | $0.3332_{(0.0047)}$ |
| TriviaQA (Joshi et al., 2017) | 1-shot | $0.0370_{(0.0014)}$ | $0.0315_{(0.0013)}$ | $0.0349_{(0.0014)}$ | $0.0298_{(0.0013)}$ |
| COPA (Wang et al., 2019) | 5-shot | $0.67_{(0.0473)}$ | $0.68_{(0.0469)}$ | $0.69_{(0.0465)}$ | $0.69_{(0.0465)}$ |
| MultiRC (Wang et al., 2019) | 5-shot | $0.5474_{(0.0071)}$ | $0.5622_{(0.0071)}$ | $0.5591_{(0.0071)}$ | $0.5680_{(0.0071)}$ |
| OpenBookQA (Mihaylov et al., 2018) | 5-shot | $0.284_{(0.0202)}$ | $0.278_{(0.0201)}$ | $0.278_{(0.0201)}$ | $0.280_{(0.0201)}$ |
| PiQA (Bisk et al., 2020) | 5-shot | $0.6458_{(0.0112)}$ | $0.6468_{(0.0112)}$ | $0.6436_{(0.0112)}$ | $0.6392_{(0.0112)}$ |
| **Average** ($\uparrow$) | | 0.4363 | **0.4383** | 0.4380 | **0.4399** |

Table 12: **Generalization performance of 774M models trained on SlimPajama-6B-DoReMi.** All models are pretrained under a $\sim$1E18 FLOPs budget. We report the best observed accuracy$_{(\text{standard error})}$ or exact match if provided, during training. **Highlighted** values indicate the best performance.

| Benchmark | Method (774M) | | | | |
|---|---|---|---|---|---|
| | # Shots | ESLM-CVaR-loss | ESLM-VaR-entropy | CLM | Rho-1 |
| ARC-E (Clark et al., 2018) | 0-shot | $0.4132_{(0.0101)}$ | $0.4158_{(0.0101)}$ | $0.4128_{(0.0101)}$ | $0.4141_{(0.0101)}$ |
| LAMBADA (Paperno et al., 2016) | 5-shot | $0.2437_{(0.006)}$ | $0.2400_{(0.006)}$ | $0.2124_{(0.0057)}$ | $0.2229_{(0.0058)}$ |
| SciQ (Welbl et al., 2017) | 5-shot | $0.799_{(0.0127)}$ | $0.801_{(0.0126)}$ | $0.78_{(0.0131)}$ | $0.8_{(0.0127)}$ |
| HellaSwag (Zellers et al., 2019) | 5-shot | $0.3417_{(0.0047)}$ | $0.3383_{(0.0047)}$ | $0.3366_{(0.0047)}$ | $0.3382_{(0.0047)}$ |
| TriviaQA (Joshi et al., 2017) | 1-shot | $0.0457_{(0.0016)}$ | $0.0470_{(0.0016)}$ | $0.0412_{(0.0015)}$ | $0.0388_{(0.0014)}$ |
| COPA (Wang et al., 2019) | 5-shot | $0.71_{(0.0456)}$ | $0.69_{(0.0465)}$ | $0.68_{(0.0469)}$ | $0.67_{(0.0473)}$ |
| MultiRC (Wang et al., 2019) | 5-shot | $0.5435_{(0.0072)}$ | $0.5602_{(0.0071)}$ | $0.5680_{(0.0071)}$ | $0.5470_{(0.0071)}$ |
| OpenBookQA (Mihaylov et al., 2018) | 5-shot | $0.288_{(0.0203)}$ | $0.294_{(0.0204)}$ | $0.28_{(0.0201)}$ | $0.284_{(0.0202)}$ |
| PiQA (Bisk et al., 2020) | 5-shot | $0.6360_{(0.0112)}$ | $0.6338_{(0.0112)}$ | $0.6430_{(0.0112)}$ | $0.6289_{(0.0113)}$ |
| **Average** ($\uparrow$) | | **0.4467** | **0.4466** | 0.4393 | 0.4382 |

E.3 ABLATION FOR INSTANCE SELECTION VS TOKEN-LEVEL LOSS SHAPING

In Figure 11, we decompose the FLOPs savings of ESLM into contributions from the proxy instance-selection phase and the token-level loss-shaping phase on a 124M model trained on the SlimPajama-6B-Unif mixture. The instance selection phase provides the primary reduction by discarding low-value sequences early, while token-level shaping adds complementary savings by pruning redundant gradient updates within the retained sequences. Together, these two components improve pretraining efficiency over standard CLM training without degrading learning convergence.

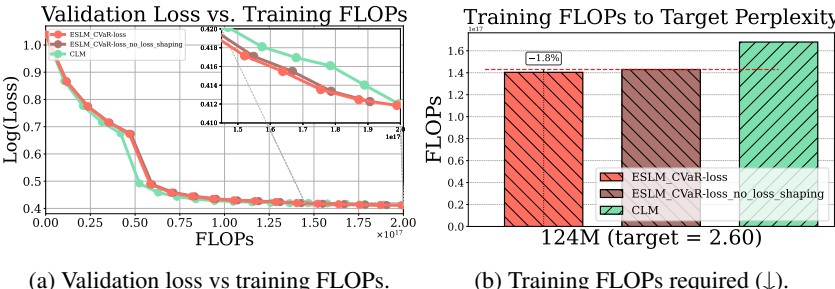

(a) Validation loss vs training FLOPs.          (b) Training FLOPs required (↓).

Figure 11: **Ablation on instance selection vs token-level loss shaping.** Applying token-level loss shaping further helps in compute-efficient convergence to the target validation loss with fewer FLOPs compared to the instance-only selection version.

# F ESLM TOKEN SELECTION ANALYSIS

To better understand the behavior of ESLM, we conduct a qualitative analysis of token selection during pretraining. Specifically, we compare the selection patterns of two ESLM variants on the same SlimPajama-6B validation sequences. Figures 13-14 present examples where highlighted tokens represent those selected for backpropagation by 124M models using a fixed confidence level $\alpha = 0.1$ (i.e., top 90% high-risk tokens retained in the objective). In Figures 15-16, we further show selected tokens by 774M models under $\alpha = 0.2$ (i.e., top 80% high-risk tokens retained). We observe that both variants prioritize rare or informative tokens—such as named entities, foreign words, and domain-specific phrases—but differ in the nature of the signals they capture:

- ESLM-VaR-entropy emphasizes tokens associated with high predictive uncertainty, often selecting structurally or semantically transitional words, including common function words (e.g., "the", "and", "of"), punctuation, and formatting artifacts when they appear in unpredictable or shifting contexts. For instance, in the second passage (Figure 15), the model selects not only semantically meaningful words such as "anxiety-inducing", "consumer-driven", but also emphasizes "the", ".", and "with" in contexts where uncertainty over their grammatical role or continuation is high.

- ESLM-CVaR-loss instead selects tokens that incur high training loss—typically semantically complex or underfit tokens. In the first passage (Figure 16), we observe selection of technical terms such as "alkanes", "aminated"; but tend to avoid repeating tokens such as "eq". This variant tends to avoid punctuation and common syntactic tokens unless they contribute directly to high loss.

Figure 12 further illustrates the frequency of top-20 selected tokens by 774M ESLM models, from the validation examples given in Figures 15-16. The results reveal that as we allow for more tokens to be selected ($\alpha$ decreases), ESLM-VaR-entropy selects syntactically ambiguous tokens such as punctuations more than ESLM-CVaR-loss, reflecting its sensitivity to positional and contextual ambiguity, even in high-frequency tokens.

Crucially, this overall analysis also highlights the strength of token-level selection: ESLM captures the informativeness within sequences, in contrast to instance-level methods such as GREATS that filter entire examples. As a result, ESLM preserves valuable learning signals that would otherwise be discarded, offering a more fine-grained and efficient form of selective pretraining.

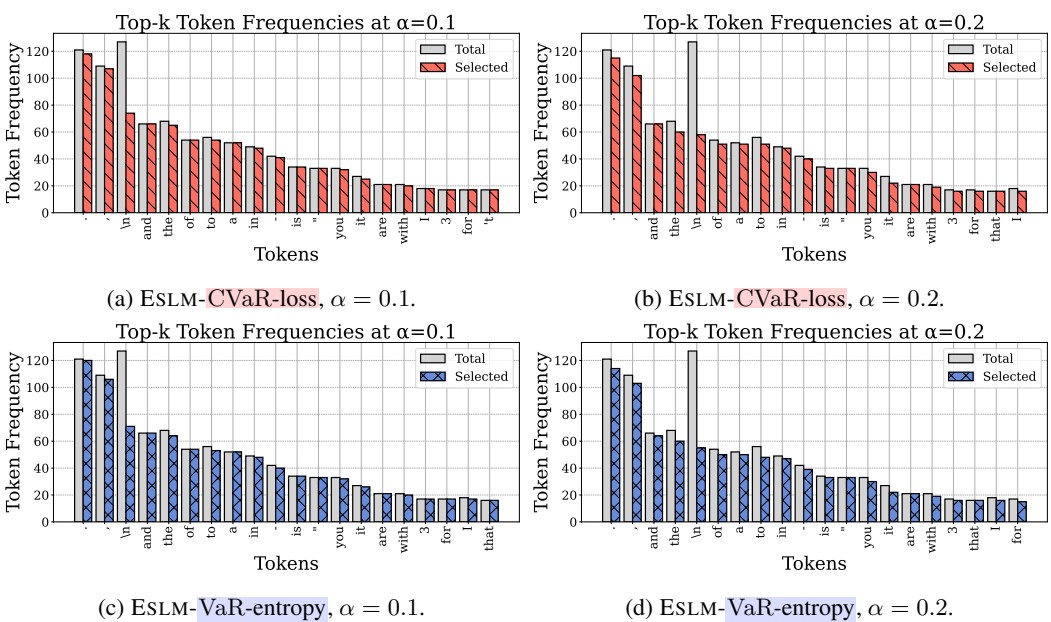

(a) ESLM-CVaR-loss, $\alpha = 0.1$.

(b) ESLM-CVaR-loss, $\alpha = 0.2$.

(c) ESLM-VaR-entropy, $\alpha = 0.1$.

(d) ESLM-VaR-entropy, $\alpha = 0.2$.

Figure 12: **Frequency of top-20 tokens selected by ESLM (774M) variants from validation sequences shown in Figures 15-16.** As $\alpha$ decreases (more tokens are selected from the batch), the ESLM-VaR-entropy emphasizes punctuation tokens more than ESLM-CVaR-loss, reflecting its sensitivity to positional and contextual ambiguity, even in high-frequency tokens.

Example 1 from domain: cc

ens, Lon Chaney Jr. Shiinomi Gakuen / The Shiinomi School (1955) Hiroshi Shimizu, KyÃƒÂ´ko Kagawa, Yukiko Shimazaki, JÃƒÂ»kichi Uno, Drama Oliver Twist (1982) Clive Donner, George C. Scott, Tim Curry, Michael Hordern, Crime, Drama Cobra Woman (1944) Robert Siodmak, Maria Montez, Jon Hall, Sabu Imburnal (2008) Sherad Anthony Sanchez, Brian Monterola, Jelieta Mariveles-Ruca, Allen Lumanog Nemuri Kyoshiro 13: The Full Moon Swordsman (1969) Kazuo Mori, Hiroki Matsukata, Tomomi SatÃƒÂ´, Sanae Nakahara, Action, Drama Shake Hands with the Devil (1959) Michael Anderson, James Cagney, Don Murray, Dana Wynter Sopyonje (1993) Kwon-taek Im, Myung-gon Kim, Jung-hae Oh, Kyu-chul Kim The American Soldier (1970) Rainer Werner Fassbinder, Karl Scheydt, Elga Sorbas, Jan George, Drama Mikey and Nicky (1976) Elaine May, Peter Falk, John Cassavetes, Ned Beatty MalÃƒÂ¡ morskÃƒÂ¡ vÃƒÂ¡la / The Little Mermaid (1976) Karel Kachyna, Miroslava SafrÃƒÂ¡nkovÃƒÂ¡, Radovan LukavskÃƒÂ½, Petr Svojtka, Family, Drama, Romance Strategic Air Command (1955) Anthony Mann, James Stewart, June Allyson, Frank Lovejoy, Action, Drama, War De Passagem / Passing By (2003) Ricardo Elias, Lohan BrandÃƒÂ£o, Thiago de Mello, Wilma de Souza, Drama The Citadel (1938) King Vidor, Robert Donat, Rosalind Russell, Ralph Richardson Ladies in Retirement (1941) Charles Vidor, Ida Lupino, Louis Hayward, Evelyn Keyes Madame SatÃƒÂ£ (2002) Karim AÃƒÂ¯nouz, LÃƒÂ¡zaro Ramos, Marcelia Cartaxo, Flavio Bauraqui, Biography, Crime, Drama Youth in Fury (1960) Masahiro Shinoda, Shin'ichirÃƒÂ´ Mikami, Shima Iwashita, Kayoko Honoo Night Plane from Chungking (1943) Ralph Murphy, Robert Preston, Ellen Drew, Otto Kruger, Action, Drama, Romance, War The Gay Falcon (1941) Irving Reis, George Sanders, Wendy Barrie, Allen Jenkins, Crime, Drama, Mystery, Romance Hndzan / Sour Grape (1974) Bagrat Oganesyan, A. Isahakyan, Sos Sargsyan, H. Azizyan, Drama Muri shinjÃƒÂ»: Nihon no natsu / Japanese Summer: Double Suicide (1967) Nagisa ÃƒÂ¶shima, Keiko Sakurai, Kei SatÃƒÂ´, Tetsuo Ashida, Crime, Drama Lady Oscar (1979) Jacques Demy, Catriona MacColl, Barry Stokes, Christine BÃƒÂ¶hm, Drama, History, Romance The Trail of 98 (1928) Clarence Brown, Dolores del Rio, Ralph Forbes, Karl Dane The Saint Meets the Tiger (1943) Paul L. Stein, Hugh Sinclair, Jean Gillie, Gordon McLeod, Crime, Drama, Mystery The Quiet Duel (1949) Akira Kurosawa, ToshirÃƒÂ´ Mifune, Takashi Shimura, Miki SanjÃƒÂ´, Drama Beyond the Sea (2004) Kevin Spacey, Kate Bosworth, John Goodman<|endoftext|>"This Moment Tests the Character of the Nation": Rep. Barbara Lee Rejects Anti-Refugee Efforts StoryNovember 18, 2015 Watch Full Show Watch Full ShowNext Story Media Options Democratic congresswoman from California and former chair of the Congressional Black Caucus. House Speaker Paul Ryan and Senate Majority Leader Mitch McConnell have called for a "pause" in the U.S. program accepting Syrian refugees. Meanwhile, governors of at least 27 U.S. states have said they will not accept Syrian refugees. We speak to California Democratic Rep. Barbara Lee. StoryJan 19, 2018As Shutdown Looms over Immigration, Trump's Rejection of Refugees Could Have Global Domino Effect AMY GOODMAN: As we turn now to Washington, D.C., as House Speaker Paul Ryan and Senate Majority Leader Mitch McConnell are calling for a pause in the U.S. program accepting Syrian refugees, I want to bring into the conversation Congressmember Barbara Lee of California. Your response to the crackdown? Now, 27 governors are saying they will not accept Syrian refugees. In fact, your theory, Peter Bouckaert, around

---

Example 2 from domain: book

out. Well, they weren't getting away with it. There had to be a confrontation. I got to my feet and marched toward the cabin. It was showdown time. Cards on the table, Gagliano! What's your game, atheist? Let's have the truth! I kicked open the cabin door and got a surprise. Sitting there drinking wine was my father. "You raised in a barn?" I closed the door carefully. "Where you been?" "Looking for you," I said. "Where _you_ been?" "Right here." "All the time?" "All the time." "Didn't you hear me calling?" "When?" It was useless to ask any more questions. I sat down and he poured me a little wine. "Eat something," he said, pushing the bread and cheese across the table. "What's hemorrhoids?" He told me, and I had to push the food away. "You're too young for hemorrhoids." "Not me. That woman." "She's got her troubles." He rolled some wine in his cheeks, staring thoughtfully. His eyes seemed dipped in blood. "Your mother's a wonderful woman," he said. I just looked at him. "Finest woman in the world." He stood up, lurching, and drifted heavily to the door and outside. I went to the door. He sat on a log a few feet away, talking to himself. "An angel," he said. Though the twilight was still warm, I put some logs in the stove and stretched out on the couch. Leaning on an elbow I watched my father through the open door. He was like a statue, chin in both hands. It was very quiet but beyond the silence you heard the uproar out there, bullfrogs croaking, birds and crickets singing, bugs buzzing, and the trees sighing in the wind. The crackling fire splashed the ceiling with wild shadows and filled the cabin with warmth. # ELEVEN It felt like midnight when I wakened. Someone had slipped off my jeans and shoes and laid a blanket over me. Shafts of moonlight poured through the windows. The fire was a mound of ashes in the stove. The other two beds were not occupied. I was alone. I put on my shoes and jeans and went outside. The moon was gigantic. From the direction of the mine I heard Frank Gagliano's drunken gravel laughter, then the voice of Rhoda Pruitt, then a roar from my father. I told myself not to go up there, to stay in the cabin, to leave them alone, but I would not listen to myself, and the presence of evil coming from there drew me up the trail, running eagerly on tiptoe enchanted by the sense of evil. They did not hear me, nor even the thunder of my heart, nor did they even see me in the frenzy of their cleaving together, grunting and sucking and squirming in the naked heavy slithering of arms and legs, caught up like a ball of squirming white snakes, bodywhite under the moon, grinding on a blanket all knotted together with them, clawing, gasping, groaning. Then I saw my father's face. It was the face of the devil on the door. I turned and ran. I ran to the cabin. I was cold, shivering. I threw wood into the fire. I shuddered, wrapped in a blanket by the fire, teeth clack, clack. Then I was thirsty, drink anything, the wine! I drank and drank. Shivering, hungry, famished. But not _their_ cheese, their hemorrhoid cheese, _their_ bread. I found the box with the sandwiches my mother had made for me, and I ate, and it was good in my mouth, sweet and good, but I shivered all the same, the blanket around my shoulders, their fire burning in my face. Then I discovered the bottle she had placed there, wrapped in a cloth, a pint of holy water. She had written upon it, written: "Holy water. Use as needed." Now I knew it, now I would do it. I went up there, running, with the bottle of holy water, a fool with holy water, I knew it, I knew I was a fool, but I didn't care. They had to know I was coming. It was only fair to let them know, they were entitled to that. I yelled, "Holy water!" I ran, yelling, "Holy water!" "Holy water on its way!" "Here comes the holy water!"

---

Figure 13: Example inputs from SlimPajama-6B-Unif mixture showing the **selected tokens** by Eslm-VaR-entropy (**124M**) with $\alpha = 0.1$. [*Note: These examples are drawn from public datasets (Soboleva et al., 2023) and may contain intense language, political references, or mature content. These excerpts are included solely for the purpose of analyzing model behavior.*]

**Example 1 from domain: cc**

ens, Lon Chaney Jr. Shiinomi Gakuen / The Shiinomi School (1955) Hiroshi Shimizu, KyÃƒÂ´ko Kagawa, Yukiko Shimazaki, JÃƒÂ»kichi Uno, Drama Oliver Twist (1982) Clive Donner, George C. Scott, Tim Curry, Michael Hordern, Crime, Drama Cobra Woman (1944) Robert Siodmak, Maria Montez, Jon Hall, Sabu Imburnal (2008) Sherad Anthony Sanchez, Brian Monterola, Jelieta Mariveles-Ruca, Allen Lumanog Nemuri Kyoshiro 13: The Full Moon Swordsman (1969) Kazuo Mori, Hiroki Matsukata, Tomomi SatÃƒÂ´, Sanae Nakahara, Action, Drama Shake Hands with the Devil (1959) Michael Anderson, James Cagney, Don Murray, Dana Wynter Sopyonje (1993) Kwon-taek Im, Myung-gon Kim, Jung-hae Oh, Kyu-chul Kim The American Soldier (1970) Rainer Werner Fassbinder, Karl Scheydt, Elga Sorbas, Jan George, Drama Mikey and Nicky (1976) Elaine May, Peter Falk, John Cassavetes, Ned Beatty MalÃƒÂ¡ morskÃƒÂ¡ vÃƒÂ­la / The Little Mermaid (1976) Karel Kachyna, Miroslava SafrÃƒÂ¡nkovÃƒÂ¡, Radovan LukavskÃƒÂ½, Petr Svojtka, Family, Drama, Romance Strategic Air Command (1955) Anthony Mann, James Stewart, June Allyson, Frank Lovejoy, Action, Drama, War De Passagem / Passing By (2003) Ricardo Elias, Lohan BrandÃƒÂ£o, Thiago de Mello, Wilma de Souza, Drama The Citadel (1938) King Vidor, Robert Donat, Rosalind Russell, Ralph Richardson Ladies in Retirement (1941) Charles Vidor, Ida Lupino, Louis Hayward, Evelyn Keyes Madame SatÃƒÂ£ (2002) Karim AÃƒÂ¯nouz, LÃƒÂ¡zaro Ramos, Marcelia Cartaxo, Flavio Bauraqui, Biography, Crime, Drama Youth in Fury (1960) Masahiro Shinoda, Shin'ichirÃƒÂ´ Mikami, Shima Iwashita, Kayoko Honoo Night Plane from Chungking (1943) Ralph Murphy, Robert Preston, Ellen Drew, Otto Kruger, Action, Drama, Romance, War The Gay Falcon (1941) Irving Reis, George Sanders, Wendy Barrie, Allen Jenkins, Crime, Drama, Mystery, Romance Hndzan / Sour Grape (1974) Bagrat Oganesyan, A. Isahakyan, Sos Sargsyan, H. Azizyan, Drama Muri shinjÃƒÂ»: Nihon no natsu / Japanese Summer: Double Suicide (1967) Nagisa ÃƒÂ¶shima, Keiko Sakurai, Kei SatÃƒÂ´, Tetsuo Ashida, Crime, Drama Lady Oscar (1979) Jacques Demy, Catriona MacColl, Barry Stokes, Christine BÃƒÂ¶hm, Drama, History, Romance The Trail of 98 (1928) Clarence Brown, Dolores del Rio, Ralph Forbes, Karl Dane The Saint Meets the Tiger (1943) Paul L. Stein, Hugh Sinclair, Jean Gillie, Gordon McLeod, Crime, Drama, Mystery The Quiet Duel (1949) Akira Kurosawa, ToshirÃƒÂ´ Mifune, Takashi Shimura, Miki SanjÃƒÂ´, Drama Beyond the Sea (2004) Kevin Spacey, Kate Bosworth, John Goodman<|endoftext|>"This Moment Tests the Character of the Nation": Rep. Barbara Lee Rejects Anti-Refugee Efforts StoryNovember 18, 2015 Watch Full Show Watch Full ShowNext Story Media Options Democratic congresswoman from California and former chair of the Congressional Black Caucus. House Speaker Paul Ryan and Senate Majority Leader Mitch McConnell have called for a "pause" in the U.S. program accepting Syrian refugees. Meanwhile, governors of at least 27 U.S. states have said they will not accept Syrian refugees. We speak to California Democratic Rep. Barbara Lee. StoryJan 19, 2018As Shutdown Looms over Immigration, Trump's Rejection of Refugees Could Have Global Domino Effect AMY GOODMAN: As we turn now to Washington, D.C., as House Speaker Paul Ryan and Senate Majority Leader Mitch McConnell are calling for a pause in the U.S. program accepting Syrian refugees, I want to bring into the conversation Congressmember Barbara Lee of California. Your response to the crackdown? Now, 27 governors are saying they will not accept Syrian refugees. In fact, your theory, Peter Bouckaert, around

**Example 2 from domain: book**

out. Well, they weren't getting away with it. There had to be a confrontation. I got to my feet and marched toward the cabin. It was showdown time. Cards on the table, Gagliano! What's your game, atheist? Let's have the truth! I kicked open the cabin door and got a surprise. Sitting there drinking wine was my father. "You raised in a barn?" I closed the door carefully. "Where you been?" "Looking for you," I said. "Where _you_ been?" "Right here." "All the time?" "All the time." "Didn't you hear me calling?" "When?" It was useless to ask any more questions. I sat down and he poured me a little wine. "Eat something," he said, pushing the bread and cheese across the table. "What's hemorrhoids?" He told me, and I had to push the food away. "You're too young for hemorrhoids." "Not me. That woman." "She's got her troubles." He rolled some wine in his cheeks, staring thoughtfully. His eyes seemed dipped in blood. "Your mother's a wonderful woman," he said. I just looked at him. "Finest woman in the world." He stood up, lurching, and drifted heavily to the door and outside. I went to the door. He sat on a log a few feet away, talking to himself. "An angel," he said. Though the twilight was still warm, I put some logs in the stove and stretched out on the couch. Leaning on an elbow I watched my father through the open door. He was like a statue, chin in both hands. It was very quiet but beyond the silence you heard the uproar out there, bullfrogs croaking, birds and crickets singing, bugs buzzing, and the trees sighing in the wind. The crackling fire splashed the ceiling with wild shadows and filled the cabin with warmth. # ELEVEN It felt like midnight when I wakened. Someone had slipped off my jeans and shoes and laid a blanket over me. Shafts of moonlight poured through the windows. The fire was a mound of ashes in the stove. The other two beds were not occupied. I was alone. I put on my shoes and jeans and went outside. The moon was gigantic. From the direction of the mine I heard Frank Gagliano's drunken gravel laughter, then the voice of Rhoda Pruitt, then a roar from my father. I told myself not to go up there, to stay in the cabin, to leave them alone, but I would not listen to myself, and the presence of evil coming from there drew me up the trail, running eagerly on tiptoe enchanted by the sense of evil. They did not hear me, nor even the thunder of my heart, nor did they even see me in the frenzy of their cleaving together, grunting and sucking and squirming in the naked heavy slithering of arms and legs, caught up like a ball of squirming white snakes, bodywhite under the moon, grinding on a blanket all knotted together with them, clawing, gasping, groaning. Then I saw my father's face. It was the face of the devil on the door. I turned and ran. I ran to the cabin. I was cold, shivering. I threw wood into the fire. I shuddered, wrapped in a blanket by the fire, teeth clack, clack. Then I was thirsty, drink anything, the wine! I drank and drank. Shivering, hungry, famished. But not _their_ cheese, their hemorrhoid cheese, _their_ bread. I found the box with the sandwiches my mother had made for me, and I ate, and it was good in my mouth, sweet and good, but I shivered all the same, the blanket around my shoulders, their fire burning in my face. Then I discovered the bottle she had placed there, wrapped in a cloth, a pint of holy water. She had written upon it, written: "Holy water. Use as needed." Now I knew it, now I would do it. I went up there, running, with the bottle of holy water, a fool with holy water, I knew it, I knew I was a fool, but I didn't care. They had to know I was coming. It was only fair to let them know, they were entitled to that. I yelled, "Holy water!" I ran, yelling, "Holy water!" "Holy water on its way!" "Here comes the holy water!"

Figure 14: Example inputs from SlimPajama-6B-Unif mixture showing the **selected tokens** by Eslm-CVaR-loss (**124M**) with $\alpha = 0.1$. [*Note: These examples are drawn from public datasets (Soboleva et al., 2023) and may contain intense language, political references, or mature content. These excerpts are included solely for the purpose of analyzing model behavior.*]

**Example 1 from domain: cc**

hydrochloric acid in methylene chloride-water, followed by separation of the organic phase, drying, and storage in solution at 0-5 Å,Â°C or below.6,7 Analysis of Reagent Purity: determination of positive chlorine can be carried out iodometrically.6,7 Handling, Storage, and Precautions: toxic and may explode, especially on heating or when concentrated. Dilute, cold solutions of NCl3 in various organic solvents are stable for several days.6 Store under inert atmosphere. Use only behind a safety shield in an efficient fume hood. Amination of Aromatics. The reaction of benzene and derivatives with NCl3 and Aluminum Chloride in organic solvents can be a useful preparation for meta-substituted amines. However, yields are only moderate, and mixtures of isomers are often obtained. Arenes include mono-7-9 and dialkylbenzenes,9,10 halobenzenes,11 biphenyl, and naphthalene.1 With trichloroamine/AlCl3, the conversions of toluene to m-toluidine (eq 1) and of 1,3-dimethylbenzene to 3,5-dimethylaniline (eq 2) in moderate yields have been observed. An addition-elimination mechanism involving a chloroarenium intermediate has been proposed for the amination reactions (eq 3).10 Amination of halobenzenes and halotoluenes with trichloramine/AlCl3 proceeds by two competing processes in moderate yields.11 For example, fluorobenzene gives predominantly m-fluoroaniline and p-chloroaniline (eq 4). It has been proposed that the former is produced by a substitution (addition-elimination) mechanism, while the latter is formed by a pathway involving nucleophilic displacement of halide in a chloroarenium cation by a nitrogen containing nucleophile (eq 5). Amination of biphenyl gives 3-aminobiphenyl (eq 6) and amination of naphthalene gives a mixture of 1- and 2-amino derivatives in low yields.1 Amination of Alkanes. The trichloramine/AlCl3 system has also been used for the amination of monocyclic12,13 bicyclic,13,14 and tricyclic,15 alkanes. C5-C8 cycloalkanes and their mono- and dimethyl derivatives are aminated in good yields.13 Methylcyclohexane12,16 and methylcyclopentane13 are converted to 1-amino-1-methylcycloalkanes on treatment with trichloramine/AlCl3 (eq 7). Treatment of decalin and hydridane with the trichloroamine/AlCl3 system affords cis-9-aminodecalin (eq 8) and cis-8-aminohydrindane, respectively, in good yields.13 The trichloroamine/AlCl3 amination route provides a simple one-step method of obtaining aminoadamantanes in high yield (eq 9).3,15 Diamantane17 can also be efficiently aminated in this fashion. When hydrocarbons which do not contain a tertiary hydrogen are subjected to reaction with NCl3/AlCl3, cationic rearrangements and fragmentations are observed.18 Amination of Alkyl-Substituted Aromatics. Various monoalkyl substituted arenes have been aminated on the alkyl side chain to form t-benzylamines in the system trichloroamine/AlCl3/t-butyl bromide (an efficient additive).19-21 p-Alkyl and p-haloisopropylbenzenes give the corresponding aminated products in high yields (eq 10). Tertiary Amines from Chlorides. When simple tertiary alkyl chlorides are exposed to NCl3 and AlCl3 in methylene chloride at -10 Å,Â°C, varying yields of the corresponding amines can be obtained. t-Pentyl chloride under these conditions provides t-pentylamine in 82% yield (eq 11), while t-octylamine is obtained in 35% yield from the corresponding chloride.2 Primary and secondary halides give isomeric amines resulting from skeletal rearrangement, as well as aziridines (eqs 12-14). Mechanistic details of this reaction have been reported.22 Vicinal Dichlorides from Alkenes. The reaction of NCl3 with a variety of mono- and disubstituted alkenes, both cyclic and acyclic, aff

**Example 2 from domain: book**

on the city's pulse, who is doing more, who is using their time to their maximum. And it can be anxiety-inducing, even when you've actively chosen not to do something. Our obsession with the next best thing and the activities of others is a blight of our consumer-driven society, and it is felt most keenly in cities. It is up to us to quiet the voice inside that asks why we always feel late to the party. The truth is that there will always be so much more happening in a city than you can ever spread yourself across, in person or even in awareness. We will always be surrounded by more different things we can possibly do. It is the difficulty of choice when faced with such a glut of opportunity that feels paralysing. Making decisions is scary, and yet being confident in the decisions we make is the key to so much happiness and fulfilment in life. The word 'decision' originates from the Latin _de_ and _caedere_ , meaning to cut off â¢Â¢Â¿Â¬Â literally slaying your options. It's learning when and what to opt in and out of that really matters, though. Have confidence in your choices: make sure that they reflect who you are, and what you enjoy. Don't succumb to peer pressure, or let yourself become a wingman in someone else's experience of city life. And don't end up doing nothing because you couldn't decide what to do. Planning ahead is a useful strategy in combating FOMO. Set dates to do things, book tickets for shows, concerts and tables at restaurants. Invite others to join you. This is a simple way of ensuring you will have things in your diary to look forward to. Engineer your own fun, and take others along for the ride; we all love the friends who are organized enough to book tickets in bulk and bring everyone together. Just be mindful of scheduling sufficient space for spontaneity too. Feeling Safe It is easy to believe that cities are dangerous. We are exposed to news reports and statistics that can terrify timid souls into thinking every stranger on the sidewalk is a criminal in waiting. Clearly crime is more prevalent in cities than in rural areas, but much of this is due to the greater concentration of people. It is vital not to be intimidated into living under the covers for fear of what might happen. Feeling safe is largely a matter of common sense and vigilance: the more vigilant we are as city inhabitants, the stronger we become together as a deterrent. In general, it makes sense to keep to places where there are other people. And just as being alert to your own safety is common sense, be aware of the safety of others too. If you happen to witness an incident, act with courage but caution. We've all heard the parable of the woman who was attacked on the street in broad daylight in front of many people, but no one intervened because they assumed someone else would. Should any of us find ourselves the unfortunate victim in such a situation, a good way to attract help is to shout out to someone individually, referring to them by what they are wearing, thereby giving them ownership of the situation and responsibility to act. It is important to foster your own feelings of safety. Don't put yourself in situations where you feel unsafe. Make connections with people in your neighbourhood. Be active and alert, not passive or invisible. As a city dweller you have a responsibility to be part of a community that looks out for its other members. We are all in it together. Feeling Clean Cities are dirty. Even the more clinical, manicured Mitteleuropean or Japanese cities have cars, and pollution, and inhabitants with germs who don't wash their hands and occasionally sneeze on the back of your neck. For anyone even moderately concerned with hygiene, urban living is a constant battle the moment you leave the sanctity of your own home. The grime of pollution is tough. Blowing your nose after a journey on any underground transport system is not a pretty sight, and imagining what's in your lungs after a day out on foot or bike is enough to induce panic. Unfortunately, dirty air is a trade-off we have to accept in return for the many pleasures of city living. Short of buying a respiratory mask, there's little you can do to shield yourself from pollution. Things are looking up, though. Fewer cars on the streets means less pollution in the air, and thankfully most cities are on board with the idea that this is the way forward. Most of us can take small comfort from knowing that things are better today than they were for our forebears, who could almost chew what they inhaled. When it comes to germs, we all fall foul of the inconsistent and selfish habits of humankind. Despite all the advice to wash our hands, catch a sneeze in a tissue and so forth, all it takes is one rogue individual not playing along to ruin it

**Example 3 from domain: book**

was the dog. "He went up the beach with Jamie," Rick said. Harriet dropped down beside me. There had been conversation and laughter as we approached, but now there was silence as they froze us out. I saw that Rick and Denny were smoking pot. Harriet noticed it too. "Be careful," she cautioned. "The Sheriff patrols this beach all the time." They smiled like wise old men. "You want a joint. Dad?" Denny said. "No, thanks." "How about you, mother?" It was ridiculous and he knew better. I said, "Your Mother isn't a pot smoker, so stop being a wise guy." "This stuff is pure gold, Dad. Sure you won't try it?" "No, thanks." "It won't hurt you, man." "Listen. I smoked pot before you were born, back when you could buy a full Prince Albert tin of it for four bits." "Ah, the good old days!" he needled. 'Tell us about it." "There isn't much to tell. Pot is a mind expander for people with shriveled brains. You need it because you're a moron." "Thanks a lot." He crushed his cigarette into the sand, pulled off his shoes and socks, and trudged toward the water. Harriet looked after him with soft eyes. "That wasn't very nice," she said. I got up and went after him. He turned as I came splashing up to the creeping tide, then continued on down the beach. I caught up with him and put my arm around his shoulder. He slapped it away. "Leave me alone." "I'm sorry." "There you go, sorry again. You're always sorry after you insult somebody. You make sure you insult them first, and then you're sorry." "I try to be honest." "Honest! You're as devious as a snake, twisting and talking until you have it your way. You're the most two-faced bastard I ever saw." I was about to say I was sorry again, but I caught myself just in time. We splashed along for another fifty yards, our white feet in the thin embroidery of foam whisking across the dark sand, until we came to a skiff beached above the water line, seaweed and debris cluttered around it. He didn't want me with him, but I hung in there stubbornly as he leaned against the old boat and lit a cigarette. I didn't know what to say to him and he didn't know what to say to me. "Let's start back," I said. "I'm fed up with you, Dad." "Oh?" "I want you to stop calling me a moron. Ever since I can remember, all the way back to kindergarten, you've called me a moron. Why don't you cut it out?" "Okay." Maybe the pot did it. Maybe it was a break-through of his anger, the hot night and the curious circumstance that had brought us together at that moment. Maybe he had wanted to say it for years, but the right mood and moment had eluded him, but now he said it, and it sounded like a carefully prepared statement he had tucked away for a propitious time. "Dad, you're a lousy writer." That couldn't be my son Denny. It had to be the marijuana, just as it had been the wine with my father when I was twenty. He had bullied me on years and on Christmas Eve, hostile with wine, I had challenged him. We had fought it out in our front yard in North Sacramento, rolling in the dirt, kicking and gouging and cursing until the neighbors separated us. So it was Christmas Eve again. "I think Mother writes better than you do. I've read your novels. They're corny, sentimental cop-outs, and I'm not even talking about your screenplays." "The screenplays aren't much," I admitted. "Why did you ever become a writer, Dad? How the hell did you ever get published?" "Oh, shit. I'm not that bad! H. L. Mencken thought I was pretty good. He published me first." "You stink, Dad, you really do." _The Tyrant_ isn't a bad book. It got great reviews." "How many copies did it sell?" "Not many, but it made a pretty good movie." "Have you seen it on TV lately?" I passed that one. "Anything else?"

Figure 15: Example inputs from SlimPajama-6B-Unif mixture showing the **selected tokens** by Eslm-VaR-entropy (**774M**) with $\alpha = 0.1$. [*Note: These examples are drawn from public datasets (Soboleva et al., 2023) and may contain intense language, political references, or mature content. These excerpts are included solely for the purpose of analyzing model behavior.*]

**Example 1 from domain: cc**

hydrochloric **acid in methylene chloride-water,** followed **by separation of the organic phase, drying, and storage in solution at 0-5 Å,Â°C or below.6,7** Analysis **of Reagent Purity: determination of positive chlorine can be** carried out **iodometrically.6,7** Handling, Storage, and Precautions: **toxic and may explode, especially on heating or when concentrated.** Dilute, cold solutions of NCl3 **in various organic solvents are stable for several days.6** Store under inert atmosphere. Use only behind a safety shield in an efficient fume hood. Amination of Aromatics. The reaction of benzene and derivatives with NCl3 **and Aluminum** Chloride **in organic solvents** can **be a useful** preparation **for meta-substituted amines.** However, **yields are only moderate, and mixtures of isomers are often obtained. Arenes include mono-7-9 and dialkylbenzenes,9,10 halobenzenes,11** biphenyl, **and naphthalene.1 With trichloroamine/AlCl3, the conversions of toluene to m-toluidine (eq 1) and of 1,3-dimethylbenzene to 3,5-dimethylaniline (eq 2) in moderate yields have** been observed. **An addition-elimination mechanism** involving a chloroarenium intermediate has **been proposed for the amination reactions (eq 3).10** Amination **of halobenzenes and** halotoluenes with trichloroamine/AlCl3 **proceeds by two competing processes in moderate yields.11** For **example, fluorobenzene gives predominantly m-fluoroaniline and p-chloroaniline (eq 4). It has** been proposed **that the former is produced by a** substitution (addition-elimination) **mechanism,** while the **latter is formed by a pathway involving nucleophilic displacement of** halide **in a** chloroarenium cation **by a nitrogen containing nucleophile (eq 5). Amination of biphenyl gives 3-aminobiphenyl (eq** 6) **and amination of** naphthalene **gives a mixture of 1- and 2-amino derivatives in low yields.1** Amination **of Alkanes. The** trichloroamine/AlCl3 **system has** also **been used for the** amination **of monocyclic12,13 bicyclic,13,14 and tricyclic3,15 alkanes.** C5-C8 cycloalkanes **and their mono- and dimethyl derivatives are aminated in** good **yields.13** Methylcyclohexane12,16 **and** methylcyclopentane13 **are** converted **to 1-amino-1-methylcycloalkanes on treatment with trichloroamine/AlCl3 (eq 7). Treatment of** decalin **and** hydrindane **with the** trichloroamine/AlCl3 **system affords cis-9-aminodecalin (eq 8) and cis-8-aminohydrindane,** respectively, **in** good **yields.13** The **trichloroamine/AlCl3** amination route provides a simple one-step method of obtaining aminoadamantanes in high yield **(eq 9).3,15** Diamantane17 **can** also **be efficiently aminated in this fashion. When hydrocarbons** which do **not contain a** tertiary **hydrogen are** subjected **to** reaction **with NCl3/AlCl3, cationic rearrangements and** fragmentations are observed.18 Amination of Alkyl-Substituted Aromatics. **Various monoalkyl substituted arenes have been** aminated on **the alkyl side chain to form t-benzylamines in the system** trichloroamine/AlCl3/t-butyl bromide **(an efficient** additive).19-21 **p-Alkyl and p-haloisopropylbenzenes give the corresponding aminated products in high yields (eq 10). Tertiary Amines from** Chlorides. **When simple tertiary alkyl chlorides are exposed to NCl3 and AlCl3 in** methylene **chloride at -10 Å,Â°C, varying yields of the corresponding amines can** be **obtained.** t-Pentyl chloride under **these conditions provides t-pentylamine in 82% yield (eq 11),** while **t-octylamine is obtained in** 35% yield **from the corresponding chloride.2 Primary and secondary** halides **give** isomeric **amines resulting from skeletal** rearrangement, **as well as aziridines (eqs 12-14). Mechanistic details of this reaction** have **been reported.22 Vicinal** Dichlorides **from** Alkenes. **The** reaction **of NCl3 with a** variety **of** mono- **and** disubstituted alkenes, **both cyclic and a**cyclic, aff

**Example 2 from domain: book**

on the city's pulse, **who is doing more, who is using their time to their maximum. And it can be anxiety-inducing, even when you've actively chosen** not **to do something.** Our **obsession with the next best thing and the activities of others is a blight of our consumer-driven society, and it is felt most** keenly **in cities. It is** up **to us to quiet the voice inside that asks** why **we always feel late to the party. The truth is that there** will always **be so much more happening in a city than you can ever spread yourself across, in person or even in** awareness. **We will always be** surrounded **by more different things we can possibly** do. It **is the difficulty of choice when faced with such a glut of opportunity that feels** paralysing. **Making decisions is scary, and yet being confident in the decisions we** make **is the key to so much happiness and** fulfilment **in life. The word 'decision' originates from the** Latin _de_ **and** _caedere **, meaning to cut off** Â¢Â¢µ **literally slaying your options. It's** learning **when and what to opt in and out of that really matters, though. Have confidence in your choices: make sure that they** reflect **who you are, and** what **you enjoy. Don't** succumb **to peer pressure, or let yourself become a wingman in** someone else's experience **of city** life. **And don't** end **up doing nothing because you couldn't decide what to do.** Planning **ahead is a useful strategy in combating** FOMO. **Set dates to do things, book tickets for shows, concerts and tables at restaurants. Invite others to join you. This is a simple way of ensuring you will have things in your diary to** look forward **to. Engineer your own fun, and take others along** for **the** ride; **we all love the friends who are organized** enough **to book tickets in bulk and bring** everyone **together.** Just **be mindful of scheduling sufficient space for** spontaneity **too.** Feeling Safe **It is** easy to believe **that cities are dangerous. We are** exposed **to news reports and statistics that can** terrify **timid souls into thinking every** stranger on **the sidewalk is a criminal in waiting. Clearly crime is more prevalent in cities than in rural areas, but much of this is** due **to the greater concentration of people. It is vital** not **to be intimidated into living under the covers for fear of** what **might happen. Feeling safe is largely a matter of** common **sense and vigilance: the more vigilant we are as city** inhabitants, **the stronger we become together** as **a deterrent. In** general, **it** makes **sense to keep to places where there are** other people. **And just as being alert to your own safety is** common sense, **be** aware of **the safety of others too. If you** happen **to witness an incident, act with courage but caution. We've all heard the parable of the woman who was attacked on the** street **in broad** daylight **in** front **of many people, but** no **one intervened because they assumed someone else would. Should any of us** find **ourselves the** unfortunate **victim in such a situation, a good** way **to attract help is to shout out to someone** individually, referring **to them by what they are wearing, thereby giving them** ownership of **the situation and responsibility to** act. **It is important to foster your own feelings of** safety. **Don't put yourself in situations where you feel unsafe. Make connections with people in your neighbourhood. Be active and alert, not passive or invisible. As a city** dweller **you have a** responsibility **to be** part **of a community that looks out for its other members. We are all in it together.** Feeling Clean **Cities are dirty. Even the more clinical,** manicured **Mitteleuropean or Japanese cities have cars, and pollution, and inhabitants with germs who** don't **wash their hands and occasionally sneeze on the back of your neck. For anyone even moderately concerned with** hygiene, **urban** living **is a constant battle the moment you leave the sanctity of your own home.** The grime **of pollution is tough. Blowing your nose after a journey on any underground transport system is** not **a** pretty sight, **and imagining what's in your lungs after a day out on foot or bike is enough to induce panic. Unfortunately, dirty air is a** trade-off **we have to accept in return for the many pleasures of** city living. **Short of buying a respiratory mask, there's little** you can **do to shield yourself from pollution. Things are looking up, though. Fewer cars on the streets means less pollution in the air, and** thankfully **most cities are on board with the idea that this is the way forward. Most of us can take small comfort from** knowing **that things are better today** than **they were for our** forebears, **who could almost chew what they inhaled.** When **it comes to germs, we all fall foul of the inconsistent and selfish habits of humankind. Despite all the advice to wash** our **hands, catch a** sneeze **in a tissue and so** forth, **all it** takes **is one rogue individual not playing along to** ruin it

**Example 3 from domain: book**

was the dog. **"He went up the beach with Jamie,"** Rick said. **Harriet dropped down beside me. There** had **been conversation and** laughter **as we approached, but now** there **was silence as they froze us out. I** saw **that Rick and Denny were smoking pot. Harriet noticed it too.** "Be careful," she **cautioned. "The Sheriff patrols this beach** all the **time."** They **smiled** like wise old **men.** "You want a **joint. Dad?"** Denny said. **"No, thanks."** "How about you, mother?" It **was ridiculous and he knew** better. **I said, "Your Mother isn't** a pot smoker, so stop being a **wise guy."** "This stuff is **pure gold, Dad. Sure you won't** try it?" **"No, thanks."** "It won't hurt you, man." **"Listen. I smoked pot before you were born, back when you could buy a** full Prince Albert **tin of it for four bits."** "Ah, the good **old days!"** he needled. **'Tell us about it."** "There isn't much to tell. Pot is a mind expander for people with** shriveled **brains. You need it because you're a** moron." **"Thanks a lot."** He crushed **his cigarette into the** sand, **pulled** off **his shoes and socks, and** trudged **toward the water's** edge. **Harriet looked** after **him with soft** eyes. **"That wasn't** very nice," **she** said. **I** got up **and went after him. He turned as I came** splashing up **to the creeping tide, then continued on** down **the** beach. **I caught** up **with him and put my** arm around **his shoulder. He slapped it away. "Leave me alone."** "I'm sorry." **"There you go, sorry again. You're always sorry after you insult somebody. You make sure you insult them first, and then you're sorry."** "I try to **be honest."** "Honest! You're as **devious as a snake, twisting and talking until you have it** your **way. You're the most two-faced bastard I ever** saw." **I was** about **to say I was sorry again, but I** caught **myself just in** time. **We** splashed **along for another fifty** yards, our **white feet in the thin embroidery of foam** whisking **across the dark** sand, **until we came to a skiff beached above the water line, seaweed and debris cluttered around it. He didn't want me with him, but I hung in there** stubbornly **as he leaned against the old boat and lit a cigarette. I didn't** know **what to** say **to him and he** didn't **know** what to say to **me.** "Let's start back," **I** said. **"I'm fed** up **with you, Dad."** "Oh?" **"I want you to stop calling me a** moron. **Ever since I** can **remember, all the way back to kindergarten, you've called me a** moron. **Why don't you cut it** out?" **"Okay." Maybe the pot did it. Maybe it was a break-through of his anger, the hot night and the curious circumstance that had brought** us **together at that moment. Maybe he had wanted to say it for years, but the right mood and moment had** eluded **him, but now he** said **it, and it sounded like a carefully prepared statement he had** tucked away **for a propitious** time. **"Dad, you're a lousy writer."** That **couldn't** be **my son Denny. It** had **to** be **the marijuana, just** as it **had been the wine with my father when I was twenty. He had** bullied **me for** years **and on Christmas Eve, hostile with** wine, **I had** challenged **him. We had fought it out in our front yard in North Sacramento, rolling in the dirt, kicking and** gouging **and cursing until the neighbors separated** us. **So it was Christmas Eve again.** "I think **Mother writes** better **than you** do. **I've read your novels. They're** corny, sentimental **cop-outs, and I'm not even** talking **about your** screenplays." "The screenplays **aren't** much," **I** admitted. **"Why did you ever become a writer, Dad? How** the **hell did you ever get published?" "Oh, shit. I'm not that bad! H. L.** Mencken thought **I** was pretty good. He published me first." **"You stink, Dad, you really** do." _"The **Tyrant_ isn't** a bad book. It got great reviews." **"How many copies did it** sell?" **"Not many, but it made a pretty good movie."** "Have **you seen it on TV** lately?" **I passed that one. "Anything else**?"

Figure 16: Example inputs from SlimPajama-6B-Unif mixture showing the **selected tokens** by ESLM-CVaR-loss (**774M**) with $\alpha = 0.1$. [*Note: These examples are drawn from public datasets (Soboleva et al., 2023) and may contain intense language, political references, or mature content. These excerpts are included solely for the purpose of analyzing model behavior.*]

