# OpenReview forum: "ESLM: Risk-Averse Selective Language Modeling with Hierarchical Batch Selection"
_ICLR.cc/2026/Conference — Submitted to ICLR 2026_

### Official Review · Reviewer_BFeC · 2025-10-17

**Soundness:** 3
**Presentation:** 3
**Contribution:** 3
**Rating:** 6
**Confidence:** 3

**Summary:**

This paper introduces a novel token-level training data selection algorithm that aims to boost training efficiency by focusing exclusively on high-risk tokens, thereby reducing FLOPs. The method further enhances performance through dynamic token selection ratios and by incorporating probabilistic signals for selecting tokens from larger models.

**Strengths:**

* **Clear Writing.** The paper is readable and well-structured.

* **Strong Motivation.** The motivation—to avoid spending full compute on low-value tokens and instead concentrate updates on high-risk (uncertain/high-loss) tokens—is compelling and well positioned relative to standard CLM and online data selection.

* **Extensive Experiments.** Compared to the previous version, the current experiments and analysis are broader, making the results more credible and the analysis more comprehensive.

**Weaknesses:**

* **Wall-clock time remains higher than CLM despite lower FLOPs.** While ESLM cuts training FLOPs, its end-to-end training time is still **longer than CLM** in the reported 124M setting (≈9.9–11.7 h vs. 5.3 h). The authors attribute the overhead to mismatches between selective sparsity and current accelerator kernels/tensor fusion, but this nonetheless constrains real-world applicability where time-to-train dominates. The paper acknowledges this in the limitation section.

* ESLM is a solid and practical recipe. However, beyond the engineering, the conceptual insight feels limited: Beyond the tasks discussed in this paper, how does the discussion of “high-risk” tokens provide utility or insight in broader tasks and scenarios? Moreover, given the limited applicability of the task itself, these two factors together limit a higher score.

**Questions:**

See Weaknesses.

---

> ### Author Response · Authors · 2025-11-24
> **Response to Reviewer BFeC (Part 1)**
>
> We thank the reviewer for their supportive evaluation and feedback. We appreciate your recognition of the ***novelty in the proposed*** training data selection algorithm, ***strong motivation*** & ***significance*** of token-level data selection, supported with ***clear writing*** and **extensive experiments**. We address your concerns below.
>
> ---
>
> * **\[Wall-clock time & FLOPs\]** We appreciate the reviewer acknowledging this practical aspect (which we transparently discussed in Section 6 and Appendix D3). Below we clarify in different angles why it does not diminish ESLM's contributions:
>   1. FLOPs are the right metric for **algorithmic** contribution as it shows **platform-independent** and **model-agnostic proxy** **for compute cost** in large-scale LLM training, reported in many compute-oriented studies **\[1,2\]**. Furthermore, recent efficiency works **\[3\]** in sparse training literature present theoretical FLOPs savings and discuss that time savings is reflected with hardware, frameworks, and low-level engineering optimization, which is beyond their scope.
>   2. ESLM is **significantly faster than SOTA online batch selection methods** in wall-clock time. As shown in Table 7, ESLM is **\~2× faster** than Rho-1: Rho-1 requires a high-quality reference model with expensive reference model inference; and **\~8.5-10× faster** than GREATS: GREATS needs per-sample gradient computation and high-quality validation set while also being memory inefficient. Hence, ESLM provides *significantly lightweight* data selection, further improving loss and downstream accuracy at the fixed compute budget compared to SOTA batch selection methods. In terms of compute, across model sizes (124M–1.5B), ESLM improves convergence FLOPs (e.g., \-22.5% at 350M).
>   3. Today’s hardware is optimized for dense computation, and limited to structured sparsity **\[3\]** rather than dynamic & unstructured sparsity. **Crucially, ESLM’s** contribution, which is algorithmic, **is orthogonal to system-level optimization,** as this is a hardware limitation, not the algorithmic design of ESLM. With a low-level systems optimization, ESLM’s FLOPs savings will be reflected in wall-clock time with kernel optimization availability; however, sparsity-aware hardware is an active yet challenging future research area [3], which is beyond the paper’s algorithmic scope.
>
> We emphasize that **the core contribution** **is *not a systems innovation*** but an **online hierarchical instance- and token-level batch selection algorithm**.
>
> ---

---

> ### Author Response · Authors · 2025-11-24
> **Response to Reviewer BFeC (Part 2)**
>
> * **[Conceptual insight and broader applicability of ESLM]** We thank the reviewer for recognizing ESLM as a "solid and practical recipe". The hierarchical selection and computational efficiency are the major strengths of ESLM. However, we emphasize that ESLM offers significant conceptual insight and broad utility that extends beyond the recipe, which we elaborate below.
>   1. **[Conceptual foundation]** The core conceptual insight of ESLM lies in its foundation within **distributionally robust optimization** (Section 3, lines 196-220), which moves selective training from an ad-hoc heuristic to a mathematically principled framework grounded in robust statistics [4]. ESLM formally links selective training to the CVaR optimization objective by using "high-risk" tokens/instances as a metric for training difficulty or model vulnerability. This is conceptually significant as it shows that we are explicitly optimizing the model to be **robust against the worst-case sub-distribution of the data**. This guarantees that training resources are maximally focused on reducing the risk associated with the model's most critical failure points, providing a level of theoretical grounding that distinguishes it from previous importance-weighting or heuristic sampling methods.
>      * Notably, the connection between worst-case optimization and **generalization** is well-established in robust statistics, but underexplored in language model pretraining. As we have shown in Table 1 and Figure 4, ESLM’s conceptual power of DRO-based selection provides **improved downstream accuracy** **(better generalization)**.
>      * From the **robustness** perspective, ESLM’s conceptual insight becomes even more powerful when we consider robustness to label noise and data quality variations. Since ESLM focuses on the *distribution of risk*, it naturally learns to manage the worst-case sub-distribution more effectively than standard CLM, enhancing the model's distributional robustness to noisy or adversarial examples.To support this further, we conducted an **additional experiment** under 5% label noise on the SlimPajama-Unif dataset. As reported in **Section 5.2 (see line 469, highlighted in blue; Figure 6\) of the updated manuscript**, ESLM variants converge to the target validation loss with fewer FLOPs compared to standard CLM. The results in Figure 6 show that even under noisy supervision, ESLM variants converge faster to lower loss values while also achieving the target validation loss with fewer FLOPs compared to standard training. This further demonstrates the **utility of ESLM’s hierarchical risk-aware filtering** not only in improving compute efficiency but also enhancing robustness.
>
>   2. **[Applicability & Utility of ESLM]** We would like to clarify that we highlighted **ESLM’s broader applicability, utility and versatility** in several key areas:
>      * With **ESLM-KD** (Section 5.1.1)**,** we have shown that ESLM is applicable **for knowledge distillation**, providing compute savings while improving generalization compared to baseline knowledge distillation approaches. This demonstrated that ESLM offers a generalizable **selective training principle** applicable to different objectives.
>      * We demonstrated the **versatility** of ESLM with **different selection variants** (VaR-entropy, CVaR-loss), **multiple model scales** (124M-1.5B), **multiple data distributions** (Uniform, DoReMi)**,** evaluated on **diverse downstream benchmarks** (Section 5) and **application contexts** (pretraining, knowledge distillation).
>      * In Section 3.1 and Appendix B, we showed that **Ada-ESLM’s principle,** which is adaptive selection via CVaR feedback, provides **curriculum learning without manual scheduling**. Hence, it offers a meta-learning principle for **self-regulating training dynamics**.
>      * Note that, by being selective, ESLM also unlocks **larger batch training.** Reducing the FLOPs has practical benefits such as training larger batches under the same compute budget in memory-constrained scenarios, and lower energy consumption (especially in FLOPs-bounded scenarios).
>      * Beyond these, ESLM can be applied for fine-tuning, which we see as an interesting future use case.
>
> ---
>
> We thank the reviewer again for the feedback and for recognizing the motivation and contribution of our work. Overall, we demonstrated ESLM’s effectiveness across different model sizes and datasets, with different use cases against data-centric baselines. We are happy to expand further if any parts remain unclear.
>
> ---
>
> > [1] Hoffmann, Jordan, et al.(2022).Training Compute-optimal Large Language Models. arXiv:2203.15556
>
> > [2] Chowdhery, Aakanksha, et al.(2023).PaLM: Scaling language modeling with pathways. JMLR
>
> > [3] Lasby, Mike, et al.(2024).Dynamic Sparse Training with Structured Sparsity.ICLR
>
> > [4] Rockafellar, R. T. & Uryasev, S.(2002).Conditional value-at-risk for general loss distributions. Journal of banking & finance, 26(7)

---

### Official Review · Reviewer_D7f3 · 2025-10-27

**Soundness:** 3
**Presentation:** 2
**Contribution:** 2
**Rating:** 2
**Confidence:** 3

**Summary:**

This paper introduces the algorithm Efficient Selective Language Modeling (ESLM) to reshapes the training objective. It prioritizing high-risk tokens while eliminating redundant gradient computation. The author further conduct experiments to demonstrate the superiority of algorithm.

**Strengths:**

1. The algorithm introduction is quite detailed.
2. The ablation study in terms of different hyper-parameters are quite detailed.

**Weaknesses:**

1. The novelty is limited. The core idea $VaR$ is not firstly proposed. I'm curious its difference compared against predictive entropy and perplexity. The intuition why we need risk-aware metric is not clear.
2. The comparison between VaR-entropy and CVaR-loss is absent. In Table 1, it looks like both methods have advantages in some benchmarks but I don't know how to use them in practice.
3. In Table 11, Table 12, Table 9, we can see that the final performance also depends on the training data and model size, which challenges the robustness of the algorithm.
4. The author claim ESLM is a bilevel adversarial game between the model and a masker. I am a little confused how the author optimize the masker. In Algorithm 1 line 11, ESLM only updates the model parameters.

**Questions:**

Please check the weakness part.

---

> ### Author Response · Authors · 2025-11-24
> **Response to Reviewer D7f3 (Part 1)**
>
> We thank the reviewer for acknowledging the **detailed ESLM algorithm presentation** and the **extensive experiments** demonstrating the **superiority** of our algorithm, supported with **quite detailed ablation studies**. Below we address your concerns and hope to clarify misunderstandings:
>
> ---
>
> * **\[1: Novelty & Motivation\]** We would like to clarify that ESLM offers a **novel online-batch selection method for efficient LLM pretraining,** as also noted by **Reviewer BFeC (`This paper introduces a novel token-level training data selection algorithm that aims to boost training efficiency…`).** Further, ESLM’s risk-aware design links input selection to distributionally robust optimization theory, which we already explained in Section 3 (line 208), as also acknowledged by **Reviewers Qcw4 and BFeC ( `...focusing computational resources on high-risk, informative examples. The approach is grounded in risk-sensitive optimization….`).** Below we clarify these points further:
>   1. **\[Novelty of ESLM\]** ESLM’s novelty is in combining the risk-aware tail selection metrics (VaR, CVaR) with predictive entropy or loss measures into LLM pretraining to define an **online**, **hierarchical instance- and token**\-filtering pipeline. Although VaR is first introduced as a risk metric in Operations Research, particularly in Robust Optimization literature [2]; *ESLM prioritizes high risk inputs via principled, adaptive tail sets for efficient **LLM pretraining***. As we already discussed in **Section 4 (line 301\)**, risk aversion is **underexplored** in the context of language modeling, which further strengthens the contribution of ESLM. Furthermore, **no prior work** combines instance-level filtering (via early-exit proxy) with fine-grained token-level loss shaping, hence **ESLM’s hierarchical two-stage selection** architecture introduces a novel online batch selection for LLM pretraining.
>   2. **\[Risk-aware metric intuition\]** Instead of using a fixed cutoff, ESLM applies a **quantile-based** **threshold** on the batch score distribution. This redefines the objective as the expected loss over the high-risk subset, **directly linking selection to distributionally robust optimization** (Section 3, line 208). This tail-risk formulation gives a robustness connection to worst-case distribution shifts and statistical guarantees on the fraction of tokens seen, which pure entropy/perplexity selection lacks. Importantly, this aspect **improves generalization** behavior, as explained in Section 3, line 216 (further supported with experimental results in Section 5.1).
>      * VaR/CVaR provide a statistically grounded way to identify difficult examples, rather than using arbitrary thresholds.
>      * Moreover, as we have shown with Ada-ESLM extension (Section 3.1 and Appendix B), using a risk-aware metric and adaptively tuning the $\\alpha$ confidence level based on model training dynamics provides a token-level curriculum learning. This supports that **by being risk aware**, we can **improve generalization while maintaining high training efficiency** (Figures 5a, 7a, 7b).
>      * We have added a clarifying paragraph early in **Section 3 (line 220\) in the updated manuscript** , highlighting further why risk-aware tail mechanism is effective rather than the risk score itself.
>
> ---
>
> * **\[2: VaR-entropy and CVaR-loss comparison\]** We would like to clarify that in **Section 3, line 187**, we discussed the difference between ESLM VaR-entropy and CVaR-loss variants and quantitatively compared them in all of our experiments (**Section 5.1**). In **Appendix F**, we have also provided a qualitative analysis for token selection behavior of both variants. Practical recommendations would be the following.
>   1. As the reviewer also noted, **both ESLM variants show superior performance** against baseline methods, while achieving similar compute savings. In practice, since CVaR-loss selection emphasizes high loss inputs, including both confidently incorrect predictions and uncertain correct ones, CVaR-loss variant improves model calibration and correct confident mistakes. That is, when robustness to misprediction is critical or on domains with label noise, CVaR-loss would highlight.
>   2. On the other hand, since VaR-entropy selects tokens with high predictive uncertainty regardless of correctness, a better use case would be domains with inherent ambiguity, when diversity of learned representations matters to  explore ambiguous or underrepresented regions of the data.
>
> ---

---

> ### Author Response · Authors · 2025-11-24
> **Response to Reviewer D7f3 (Part 2)**
>
> * **\[3: Performance & robustness of ESLM\]** To clarify, the reviewer’s observation actually validates **our method's generality**. The variation across settings is *indeed expected* and positive, because:
>   1. Different datasets have different characteristics: Uniform vs DoReMi use different domain mixtures, and the downstream task performance naturally varies with domain composition. However, notably ESLM consistently **outperforms or matches baselines** across both mixtures, while (importantly) **being a significantly lightweight** online batch selection method. Note that SOTA baselines Rho-1 [3] require a high-quality reference model with expensive reference model inference, and GREATS [4] needs per-sample gradient computation and a high-quality validation set while also being memory inefficient.
>   2. Larger models exhibit different learning dynamics: It is well-established that model scale affects optimal training strategies \[1\]. However, importantly, **ESLM provides compute gains across all scales** tested (124M-1.5B) as shown in Figure 2 (e.g., 14.1% FLOPs savings with 1.5B model)\.
>
> ESLM exhibits robustness since (i) it shows **consistent efficiency gains** across all settings (Figure 2: 6.5-22.5% FLOPs savings) compared to best performing baseline, (ii) it **improves generalization** (Tables 1, 8-12), **without relying on external supervision such as reference models or curated datasets; or memory inefficient computations** as baseline batch selection methods do.
>
> ---
>
> * **[4: Bilevel adversarial game]** To clarify, as shown in Eq. 3 in Section 3, the masker computes the VaR threshold (see also VaR definition in Eq. 1, line 137), which defines the worst-case token subset. And the model updates against this adversarial selection. In the next iteration, the masker adapts, i.e., a new VaR threshold is computed based on updated model parameters. Hence, the masker is not learned iteratively, it is defined by a constrained optimization rule. This still creates adversarial dynamics without explicit adversary parameters.
>   1. Additionally, note that in **Ada-ESLM extension (in Section 3.1 and Appendix B),** the masker has a learnable parameter: confidence level $\alpha\_k$, which is dynamically updated using the CVaR feedback to control the difficulty of training.
>
> ---
>
> We thank the reviewer for their feedback. Overall, we extensively demonstrated ESLM’s effectiveness across diverse model sizes, datasets mixtures, and against state-of-the-art batch selection methods. We are happy to expand further if any parts remain unclear.
>
> ---
>
> > [1] Kaplan, Jared, et al.(2020).Scaling Laws for Neural Language Models. arXiv:2001.08361
>
> > [2] Rockafellar, R. T., & Uryasev, S. (2002). Conditional value-at-risk for general loss distributions. Journal of banking & finance, 26(7), 1443-1471.
>
> > [3] Lin, Z., et al. (2024) Rho-1: Not All Tokens Are What You Need. NeurIPS.
>
> > [4] Wang, J. T. et al. (2024). GREATS: Online Selection of High-quality Data for LLM Training in Every Iteration. NeurIPS.

---

> ### Author Response · Authors · 2025-11-27
> **Kindly consider continuing the discussion**
>
> Dear Reviewer D7f3,
>
> We would greatly appreciate if you could engage further in the discussion by sharing your thoughts on our responses. We have addressed all the concerns and revised the manuscript accordingly.
>
> * We also want to emphasize that several concerns mentioned were primarily high-level or restatements of the text. We hope the detailed explanations in the rebuttal help make the core technical contributions clearer.
>
> If you have any remaining questions, we would be happy to provide further clarification to help improve your evaluation.
>
> Thank you again for your time and effort in the review process.
>
> The authors

---

### Official Review · Reviewer_Qcw4 · 2025-10-29

**Soundness:** 3
**Presentation:** 3
**Contribution:** 2
**Rating:** 4
**Confidence:** 4

**Summary:**

ESLM introduces a hierarchical selection framework that filters training data at both the sequence and token levels, focusing computational resources on high-risk, informative examples. The approach is grounded in risk-sensitive optimization. An adaptive variant dynamically adjusts selection strictness during training, and the method extends to enable efficient knowledge distillation. Empirically, ESLM reduces training compute.

**Strengths:**

- Improving the sample efficiency during LLM pre-training warrants large potential more effective pipelines
- The hierarchical nature of ESLM seems to be quite useful to actually reduce the token count during pre-training (in contrast to previously existing approaches where instances or tokens were down-weight without actually getting removed)
- I appreciate the evaluation of the computational cost (FLOPs)

**Weaknesses:**

- I am wondering how the dynamics over time change when the objective changes towards being risk averse? It is not fully clear to me how ESLM treats extreme instances or tokens. How does the behavior of LLMs change in edge cases when tuning the risk score threshold? I am refering to extreme cases (e.g., instances that carry little information or extremely long token sequences).
- There is an instance re-weighting paper that was presented at ICLR 2025 that also uses batch statistics only and seems to have similar performance characteristics [1]. I am wondering how much benefit each stage (esp. stage 1) of ESLM adds on its own.
- The models discussed in the paper are small and the pre-training dataset of choice (SlimPajama-6B) is also not large enough to yield sufficient insights into model quality. While 1.5B parameter models are frequently used for ablation studies, these models are typically trained with 30B+ tokens [1].
- The paper discusses reproducibility in various points. I am wondering if there is a code implementation that comes with the paper and if there is a plan to open source it? A good way to make it available during the review process is an anonymous git repo.


**Sources**

[1] Dynamic Loss-Based Sample Reweighting for Improved Large Language Model Pretraining, Sow. et al., 2025

**Questions:**

Q1) How many training steps does the model need in order to be able to provide a sufficiently good signal for ESLM stage 1? I would think, there needs to be some kind of a warm up.

Q2) How would ESLM perform when using MoE models? I would assume the stage 1 proxy would not work the same way as it does for dense models.

---

> ### Author Response · Authors · 2025-11-24
> **Response to Reviewer Qcw4 (Part 1)**
>
> We thank the reviewer for their feedback. We are glad you found the ESLM approach ***useful*** for computational efficiency compared to prior work, and appreciated ESLM’s **large potential for more effective** LLM pretraining pipelines and evaluation of the **compute cost**.
> Below, we clarify your concerns and questions:
>
> ---
> * **[Dynamics and risk threshold behavior]** We address the different interpretations of your concern below:
>   * ESLM's risk-aware selection naturally handles extreme cases through its **hierarchical** design. Extreme low-risk sequences (highly predictable, low-entropy) are efficiently filtered out in the proxy phase (Stage 1\) to avoid wasting full model compute on such low information inputs. Since we obtain sequence scores via *mean* token-level statistics over non-padded tokens, length itself does not bias selection. Furthermore, since Stage 2 operates within the selected sequences, the sequences with mixed token informativeness retain their high-value tokens while masking redundant ones, ensuring compute is allocated where the learning signal is strongest.
>   * As we have shown in our **$\alpha$ ablation** (see Section 5.2, Figure 5c), lower $\\alpha$ yields broader token coverage but higher compute; whereas high $\\alpha$ value leads to more selective behavior that might exclude useful signals. Among the ablation range of $\alpha \\in \\{0.05, 0.1, 0.2, 0.3, 0.5\\}$ which includes high risk tokens, the efficiency and coverage tradeoff is effectively balanced when $\\alpha \\in \[0.1, 0.2\]$.
>   * Importantly, **Ada-ESLM** **extension** (see Section 3.1; Appendix B, Algorithm 2\) **precisely addresses** the reviewer’s concern of tuning risk score threshold, by **dynamically adjusting** **$\\alpha$ based on actual training dynamics**–particularly, using the CVaR feedback during training. Ada-ESLM provides a principled curriculum depending on the training difficulty. As shown in Figure 6, Ada-ESLM naturally starts with lower $\\alpha$ (broader coverage) and adjusting selectivity as the model improves. If the batch contains many hard instances, CVaR spikes and Ada-ESLM decreases $\\alpha$ to expand coverage. Whereas, when the model has learned most patterns and few hard tokens remain, then CVaR drops and Ada-ESLM increases $\\alpha$ to become more selective–focusing more on challenging tokens.
>
> ---
>
> * **[Comparison to reference [1] & ESLM per-stage contribution]**
>   Thank you for pointing us to a recent reference, which we have discussed in comparing instance vs token level selection in Section 3 (line 247\) of our paper. We first clarify the key distinctions and then address ESLM’s per-stage contribution.
>   * **[Comparison to [1]]** In terms of ***granularity***, [1] operates at the instance level with reweighting, while ESLM performs **hierarchical** instance-then-token selection with **removal** of inputs. This enables token-level precision that instance-only methods cannot achieve. In terms of ***mechanism design***, ESLM removes tokens and achieves true FLOPs savings through eliminated forward computation and backward gradient computation. Moreover, considering **theoretical grounding**, ESLM provides a bilevel game formulation and explicit connection to distributionally robust optimization, offering principled justification beyond empirical schemes.
>   * **[ESLM per-stage contribution]** To clarify, as we have already presented in ablation study in Appendix E.3 (Figure 10), ESLM achieves **\-14.89% FLOPs** reduction with ***Stage 1 only*** compared to CLM; whereas the savings rises to **\-16.4% FLOPs** with ESLM ***Stage 1+2 combined***, compared to CLM. This is an expected behavior. Our analysis showed that proxy phase (Stage 1\) acts as an efficient coarse instance filtering, providing the primary reduction; while token-level loss shaping (Stage 2\) provides fine-grained refinement, adding complementary savings by pruning redundant gradient updates within the retained sequences.
>
> ---
>
> * **[Model sizes and pretraining corpora shows scalability and diversity]** We clarify that, **crucially, ESLM shows scaling & diversity consistency**. That is, ESLM achieves consistent improvements across all scales tested (124M-1.5B), across different dataset mixtures (Uniform, DoReMi), and on diverse downstream tasks (9 benchmarks). This **consistency strongly suggests** the method will **scale further** and yield sufficient insights for **model quality**.
>   * Note that our experimental setup is also consistent with the standard practices in SOTA batch selection for efficiency-focused pretraining research: GREATS \[2\] and Rho-1 \[3\] are trained with \<30B tokens, GREATS further considers only the 124M model. Unlike these, we prioritized breadth (4 model sizes, 2 datasets, multiple baselines, extensive downstream evaluation) over depth at a single scale, and further showed its extension (Ada-ESLM) and use-case for knowledge distillation (ESLM-KD).

---

> ### Author Response · Authors · 2025-11-24
> **Response to Reviewer Qcw4 (Part 2)**
>
> * **[Reproducibility]** In the **supplementary material** in our updated submission, we provide the key implementation functions for ESLM. As we explain in Appendix-D6, our implementation is based on the NanoGPT and DoReMi codebases, and we provide a detailed algorithmic description of ESLM in Algorithm-1. We commit to releasing the complete implementation in a git repository in the camera-ready version.
>
> ---
>
> * **[Q1]** The advantage of ESLM is that the input selection is applied **from the start**, and instance & token selection is adjusted according to the current model state. Hence, no warm-up is applied, because:
>   * The shallow proxy (Figure 5d) captures coarse-grained patterns sufficient for instance filtering even with limited training. We show in Section 5.2 that even L=1 is effective.
>   * Even randomly initialized models exhibit relative differences in loss/entropy across examples. While absolute values might be high early in training, the ranking of examples by difficulty is informative from the start.
>   * Training curves (Figures 3a, 3c) show that there is **no initial degradation** period that would suggest insufficient signal quality.
>   * **Ada-ESLM** extension **provides implicit curriculum**, particularly addressing concerns about early training. As we have shown in Figure 6 in Appendix B, Ada-ESLM naturally starts with a lower $\\alpha$ (broader coverage) and increases selectivity as the model improves, implementing a principled curriculum without explicit warm-up.
>
> ---
>
> * **[Q2]** We thank the reviewer for the insightful question. ESLM could be integrated into MoE models, considering the following practical aspects:
>   * Proxy pass would route through active experts in the first L layers, which may result in different expert selection patterns between proxy and full model. **However,** the **coarse-grained signal** for instance filtering remains valid, as early layers capture fundamental input characteristics.
>   * Stage 2 should work seamlessly as token-level selection operates on final-layer outputs.
>
> ESLM would provide ***complementary*** **sparsity** by selecting which tokens to train on aside from expert routing in MoE, potentially amplifying efficiency gains. We think this would be an interesting future research direction.
>
> ---
>
> We thank the reviewer for their feedback. With the additional clarifications, we extensively analyzed ESLM’s effectiveness across diverse model sizes, dataset mixtures, and against state-of-the-art batch selection methods. We are happy to expand if further questions remain.
>
> ---
>
> > [1] Sow. et al. (2025) Dynamic Loss-Based Sample Reweighting for Improved Large Language Model Pretraining. ICLR.
>
> > [2] Wang, J. T. et al. (2024). GREATS: Online Selection of High-quality Data for LLM Training in Every Iteration. NeurIPS.
>
> > [3] Lin, Z., et al. (2024) Rho-1: Not All Tokens Are What You Need. NeurIPS.

---

> > ### Comment · Reviewer_Qcw4 · 2025-11-26
> >
> > Thank you for your concise response. I still disagree on the scalability claim. Rho-1 uses up to 80B tokens across models to discuss pre-training performance (esp. on the 1.1B Llama model, see section 3.3 in their paper) but I understand that such experiments are difficult/expensive to do. My other concerns have been addressed.
> > I have updated my rating accordingly.

---

### Author Response · Authors · 2025-11-24
**General Response**

We thank the reviewers for their time, and their feedback.

The reviewers found our approach to be **novel, well-motivated, technically sound, and practically relevant with comprehensive empirical evaluations**.

They highlighted that ESLM presents a novel and hierarchical framework for compute-efficient LLM pretraining (`“This paper introduces a novel token-level training data selection algorithm…”` (BFeC); “`The hierarchical nature of ESLM seems to be quite useful to actually reduce the token count during pre-training”` (Qcw4)), and appreciated its strong empirical foundation and clarity (“`The algorithm introduction is quite detailed”` (D7f3); `“Clear writing. The paper is readable and well-structured”` (BFeC)). Reviewers acknowledged the value of our risk-aware formulation and its adaptive mechanism for efficient training (`“Improving the sample efficiency during LLM pre-training warrants large potential for more effective pipelines”` (Qcw4)), and commended the comprehensive experiments and analyses supporting the method’s effectiveness (`“The ablation study in terms of different hyper-parameters are quite detailed”` (D7f3); `“...the current experiments and analysis are broader, making the results more credible and the analysis more comprehensive”` (BFeC)).

Overall, the reviews recognized ESLM as a promising and principled contribution toward compute-efficient and robust large-scale language model training.

---

In summary, the main contribution of our work is to introduce **ESLM**: an online hierarchical instance and token-level selective training method grounded in risk-aware (VaR/CVaR) selection that improves computational efficiency and generalization performance for LLM pretraining.

---

We emphasize (with detailed clarification below) that
- (i) ESLM’s risk-aware selection mechanism provides a statistically principled mechanism to identify informative tokens, connecting it to distributionally robust optimization \[1,2\] & bilevel games \[3\] which improves generalization. For this, we provide a discussion in **Section 3 in our updated manuscript (line 220\)** *\[addressing the reviewer D7f3’s comment\]*,
- (ii) FLOPs are used as a model-agnostic proxy for compute cost, consistent with the large-scale LLM training (Section 3\) and we are explicit about the gap to wall-clock time, as detailed in Appendix D3. We also provide an additional experiment in **Section 5.2 of the updated manuscript** that showcases the utility of ESLM’s hierarchical risk-aware filtering for robustness to label noise *\[addressing the reviewer BFeC’s concern\]*,
- (iii) Ada-ESLM extension which is introduced in Section 3.1 (also in Appendix B, Algorithm 2\) and our $\\alpha$ ablation study (Section 5.2, Figure 5c) clearly show how the behavior changes under varying risk score threshold *\[addressing the reviewer Qcw4’s concern\]*.

---

> [1] Duchi, J. C., & Namkoong, H. (2021). Learning models with uniform performance via distributionally robust optimization. The Annals of Statistics, 49(3), 1378-1406.

> [2] Kuhn, D., Shafiee, S., & Wiesemann, W. (2025). Distributionally robust optimization. Acta Numerica, 34, 579-804.

> [3] Curi, S., Levy, K. Y., Jegelka, S., & Krause, A. (2020). Adaptive sampling for stochastic risk-averse learning. Advances in Neural Information Processing Systems, 33, 1036-1047.

---

### Meta-Review · Area_Chair_u5rp · 2026-01-13

**Summary:**

This paper introduces Efficient Selective Language Modeling (ESLM), an online hierarchical batch selection framework for compute-efficient and robust large language model pretraining. ESLM combines instance-level filtering using risk-aware (VaR/CVaR) criteria with token-level loss shaping to prioritize high-risk tokens while eliminating redundant computation. The method is framed as a bilevel game and linked to distributionally robust optimization, with an adaptive variant (Ada-ESLM) that adjusts selectivity during training. Experiments on GPT-2–scale pretraining demonstrate significant FLOPs reduction while maintaining or improving perplexity and downstream task performance across model sizes and datasets.

**Reviewer Concerns:**

Reviewers raised concerns about the scalability claim, noting that while FLOPs are reduced, wall-clock training time remains higher than standard CLM due to hardware and kernel inefficiencies. Some reviewers questioned the conceptual novelty beyond a strong engineering contribution and asked for clearer intuition distinguishing risk-aware selection from entropy- or loss-based heuristics. One reviewer expressed confusion about the bilevel formulation and robustness claims, though these concerns were largely attributed to misunderstandings.

**Reviewer Scores:**

The authors clarified the theoretical motivation of risk-aware (VaR/CVaR) selection and its connection to distributionally robust optimization, explained the bilevel adversarial formulation, and provided extensive ablations isolating the contributions of instance- and token-level selection. They added experiments on label noise robustness, adaptive thresholding (Ada-ESLM), and comparisons across model scales, datasets, and knowledge distillation. Reviewer questions about dynamics, extreme cases, warm-up, and MoE applicability were directly addressed. Though some of the issues have been addressed, I believe this paper would benefit from going through one more round of reviews by having the authors resubmit an enhanced version with the issues addressed.

---

### Decision · Program_Chairs · 2026-01-26

Reject